# Metamodelling of a two-population spiking neural network

**Jan-Eirik W. Skaar**[1]*, **Nicolai Haug**[1], **Alexander J. Stasik**[1,2], **Gaute T. Einevoll**[1,2], **Kristin Tøndel**[1]

**1** Faculty of Science and Technology, Norwegian University of Life Sciences, Ås, Norway, **2** Department of Physics, University of Oslo, Oslo, Norway

* jan-eirik.welle.skaar@nmbu.no

**Data Availability Statement:** All relevant data are within the manuscript.

**Funding:** GTE and AJS received funding from the Research Council of Norway (DigiBrain 248828, CoBra 250128), https://www.forskningsradet.no/.

## Abstract

In computational neuroscience, hypotheses are often formulated as bottom-up mechanistic models of the systems in question, consisting of differential equations that can be numerically integrated forward in time. Candidate models can then be validated by comparison against experimental data. The model outputs of neural network models depend on both neuron parameters, connectivity parameters and other model inputs. Successful model fitting requires sufficient exploration of the model parameter space, which can be computationally demanding. Additionally, identifying degeneracy in the parameters, i.e. different combinations of parameter values that produce similar outputs, is of interest, as they define the subset of parameter values consistent with the data. In this computational study, we apply metamodels to a two-population recurrent spiking network of point-neurons, the so-called Brunel network. Metamodels are data-driven approximations to more complex models with more desirable computational properties, which can be run considerably faster than the original model. Specifically, we apply and compare two different metamodelling techniques, masked autoregressive flows (MAF) and deep Gaussian process regression (DGPR), to estimate the power spectra of two different signals; the population spiking activities and the local field potential. We find that the metamodels are able to accurately model the power spectra in the asynchronous irregular regime, and that the DGPR metamodel provides a more accurate representation of the simulator compared to the MAF metamodel. Using the metamodels, we estimate the posterior probability distributions over parameters given observed simulator outputs separately for both LFP and population spiking activities. We find that these distributions correctly identify parameter combinations that give similar model outputs, and that some parameters are significantly more constrained by observing the LFP than by observing the population spiking activities.

## Author summary

In computational neuroscience, mechanistic models are used to simulate networks of neurons. These models exhibit complex dynamics, and the parameters of the neurons and connections between neurons shape the model's behaviour. Due to the model complexity,

GTE and HEP received funding from European
Union's Horizon 2020 Framework Programme for
Research and Innovation under Grant Agreements
No. 945539 (Human Brain Project SGA3).
Simulations and analysis were performed on the
DEEP system at Jülich Supercomputing Centre
built with funding from the European Union's
Horizon 2020 Programme under Grant Agreement
754304 (DEEP-EST). The funders had no role in
study design, data collection and analysis, decision
to publish, or preparation of the manuscript.

**Competing interests:** The authors have declared
that no competing interests exist.

running the simulations and fitting the model to experimental data can be computationally demanding. In this study, we train and compare different metamodelling techniques, data-driven approximations that are much faster to run, to two different signals generated by a two-population recurrent network model, the population spiking activities and the local field potential (LFP). Further, we invert the metamodels, and demonstrate that it can reliably find the different combinations of parameters that can give rise to an observed simulation output. We compare the accuracy of the metamodels on both the forward and inverse problem, and investigate to what degree the parameters are constrained by observing the two different signals.

## 1 Introduction

Mechanistic modelling of neurons and networks of neurons based on the underlying biophysical principles is a well-established field and is an important part in bridging the scales between individual neurons and higher-level brain function [1–4]. Neuron models range in complexity from rate-based models, where spikes are not explicitly modelled, to point-neurons, where the detailed morphology of the individual neurons are collapsed into a single point, and further to multi-compartment models where the full morphology is accounted for. At the network level, even simple models exhibit complex dynamic behaviours depending on both neuron dynamics and network connectivity [5]. A question modellers face is how the network model can be parameterized in order to produce a specific behaviour, and how sensitive the behaviour is to changes in model parameter values. As mechanistic models can be expensive to run and typically have many parameters, that is, high-dimensional parameter spaces, accurately sampling the space of model behaviours is a nontrivial problem.

Mechanistic models of dynamical systems typically consist of a set of differential equations that can be integrated forward in time. The modeller may be interested in a subset of the model's behaviours, e.g. the state variables of the model, such as membrane potentials, synaptic conductances or currents, individual neuron behaviours such as spike times, or aggregated statistics such as population firing rates, local field potentials [6], or current source densities. The user specifies a set of parameters, which will determine the model's behaviour. If the model is deterministic (no random variables are involved), the output will be a unique function of its parameters and initial conditions. If the model is probabilistic (one or more random variables affect the model), the output will no longer be a unique function of the parameters and initial conditions, but will instead follow a probability distribution over possible outputs.

Conceptually, a mechanistic model can be thought of as a function $\mathcal{M}$ that takes input values and returns output values,

$$\text{output} = \mathcal{M}(\text{input}) \ .$$

If the model is stochastic, there is a distribution over possible outputs, which typically does not have a closed-form expression, since explicitly evaluating it would require integrating over all possible internal model states. *Metamodels* are data-driven statistical models that take a set of inputs and outputs generated by a mechanistic model, and learn an approximation of $\mathcal{M}$ over a restricted part of its domain [7]. The metamodel can provide either a point-estimate of the model output, or a distribution over possible outputs. In the case of stochastic models, the latter case has the advantage of providing an approximation of the intractable probability distribution over simulator outputs. The number of samples needed to achieve sufficient accuracy over the domain of interest will depend on the dimensionality of the inputs, on how the

function $\mathcal{M}$ changes with respect to the input dimensions, and on whether it has a structure that the metamodel is able to capture.

The inputs can in principle be anything that changes the model behaviour, e.g. model parameters or some other specific model input. As the volume of the parameter space increases exponentially with the number of parameters, densely sampling high-dimensional parameter spaces is very costly. Metamodels can provide a compact representation of the model over the domain, where the density of sampling required can be determined by evaluating the model on a test data set. Metamodels can also provide simpler interpretations of the model, identify the importance of individual input dimensions, and give an overview of the original model's behavioral repertoire, i.e. a statistical depiction of the possible outputs the model can produce under a range of input conditions [8]. Metamodels can be applied in either direction, from model input to model output, or vice versa, directly approximating the inverse model.

For stochastic models, the distribution over model outputs will depend on the parameter values and possibly other model inputs, giving a conditional probability $p(\mathbf{x}|\boldsymbol{\theta})$, often called the *model likelihood*, where $\mathbf{x}$ is the model output and $\boldsymbol{\theta}$ represents model parameters and other types of inputs. Since different model inputs possibly can give rise to the same model outputs, there is also a distribution over model inputs that can give rise to the same model outputs, $p(\boldsymbol{\theta}|\mathbf{x})$, often called the *posterior distribution*.

Estimating this posterior distribution by running simulations, without direct access to the model likelihood, has been the focus of work such as Approximate Bayesian computation (ABC) [9, 10], Sequential Neural Posterior Estimation (SNPE) [11, 12] and Sequential Neural Likelihood Estimation (SNLE) [13]. SNPE directly models the posterior, typically as a normalizing flow parameterized by a neural network, while SNLE models the likelihood, and infers the posterior in a separate step. The present work focuses on creating an approximation to the likelihood, as in the first step in SNLE, and can thus also be used to infer posterior distributions in the same manner. These methods can also be focused on estimating the posterior probability $p(\boldsymbol{\theta}|\mathbf{x}_0)$ for a specific observed output $\mathbf{x}_0$, and generating a local model of the posterior distribution around that point by iteratively running simulations where the posterior density is believed to be high. This approach can be viewed as a metamodel that is trained locally around a specific point of interest in the parameter space. In this case, the metamodel will not have to be as flexible as in the case where one is interested in a much larger volume of the parameter space.

Neuronal networks typically exhibit stochastic dynamics, where input currents and firing may be stochastic, and the network connectivity and initial conditions can be drawn randomly. The network activity is also nonlinearly dependent on the precise history of the activity [14]. This makes direct metamodelling of time series data challenging, as different instantiations of the same model will produce different results, which the metamodel will have to account for. Simplified mechanistic models of population rates have been developed, ranging from models based on mean-field theory [5, 14] where the number of neurons is assumed to be infinite and finite-size effects are disregarded, to ones where finite-size effects are fully accounted for [15], but where stochastic firing of neurons and full connectivity between all neurons are assumed.

In the present work, we train metamodels to approximate the distribution over the power spectra of the population spiking activities and local field potential (LFP), based on the model parameters of the Brunel network [5], comprising an excitatory and an inhibitory population of recurrently connected point-neurons. The metamodels approximate the conditional probability distribution $p(\mathbf{x}|\boldsymbol{\theta})$, where $\mathbf{x}$ is the power spectrum of either the population spiking activity or the LFP, and $\boldsymbol{\theta}$ is the biological parameters of the spiking network. The LFP, the low-frequency part of extracellular potentials, largely represents how synaptic inputs are processed

in the dendrites [6, 16, 17], so we chose to include both signals as they in a sense represent the incoming and outgoing signals for a population, and it offers the opportunity to compare the information carried by the two different signals. Fig 1 shows an overview of the modelling scheme we employ.

We train metamodels based on Gaussian process regression (GPR) [18], and compare them to a masked autoregressive flow (MAF) [19], a model often used for density estimation tasks. Both models can be trained to approximate the conditional probability densities of the simulator. Due to advances in sparse approximations to Gaussian process (GP) models, allowing them to be used on much larger data sets [20, 21], and extensions to deep models (DGPRs) [22–24], which gives them increased flexibility, GPs provide a powerful class of nonparametric regression and classification models. MAF is a type of normalizing flow, a parametric model based on making successive invertible transformations, typically based on neural networks, to a simple base distribution, typically assumed to be Gaussian. It can be trained either as a conditional distribution, where the neural networks take an additional input representing the condition, or as a simpler distribution without any conditioning. Models representing conditional distributions are particularly useful as a metamodel for systems with stochastic dynamics, since that implies that there is a distribution over possible outputs that has to be modelled. We additionally train a Hierarchical Cluster-based Partial Least Squares Regression [25]

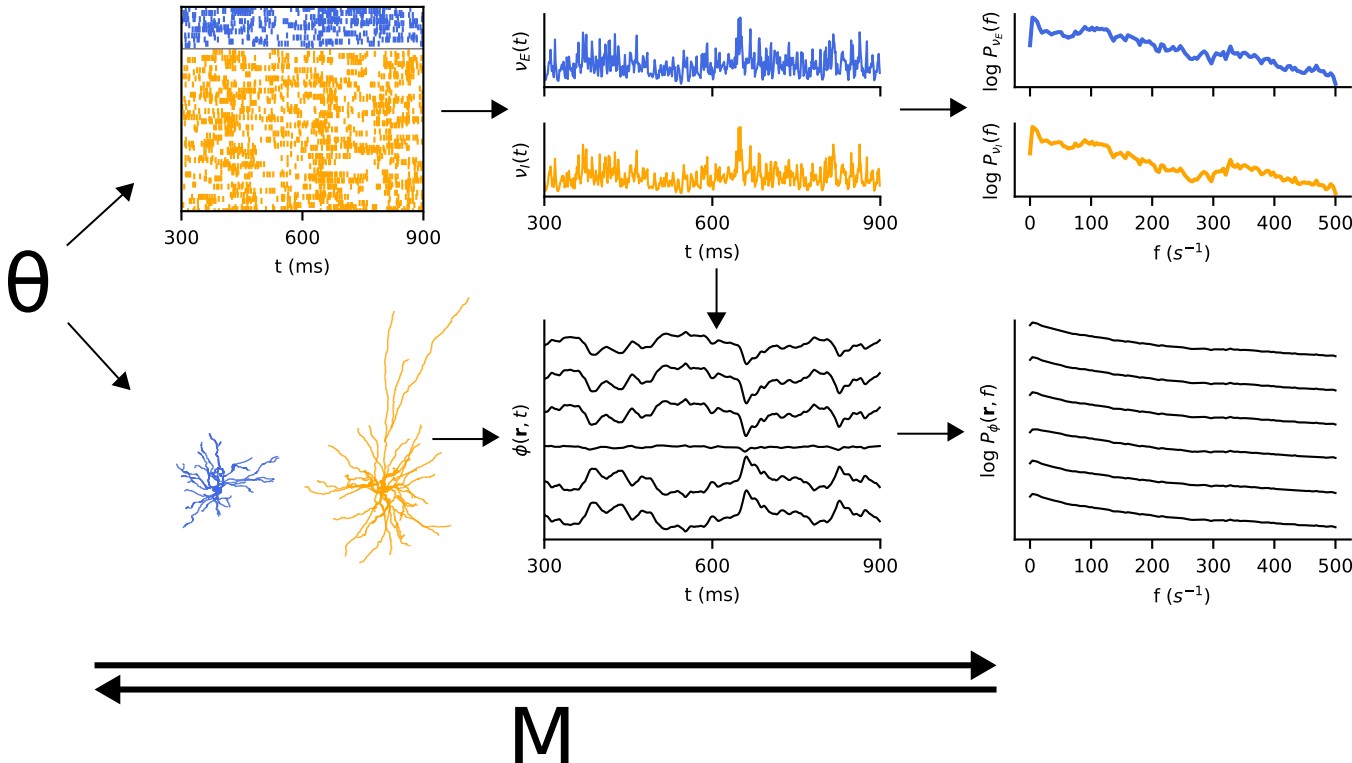

**Fig 1.** Model overview: Given a set of parameters $\boldsymbol{\theta}$, a network of point-neurons are simulated, from which the population spiking activities $v(t)$ are computed (top). With morphological neurons spatially extended in a column, the synaptic spiking activities from the point-neuron network can be 'replayed' on the morphological neurons, and the resulting local field potential $\phi(\mathbf{r}, t)$ at 6 locations is computed (bottom). Note that only one excitatory and one inhibitory neuron is shown, in simulation there is one for each point-neuron. The power density spectra $P_v(t)$ and $P_\phi(\mathbf{r}, t)$, of the population spiking activities and local field potential respectively, are also computed. The metamodel M directly models the power spectra given the parameter set, or the parameters given observed power spectra.

(HCPLSR), a model that has been used previously for metamodelling of biological systems but only gives a point-estimate of the simulator output [8, 26].

We show that the simulator can be accurately captured by metamodels trained on around 1000 example simulations, and that for the present situation, the DGPR metamodel is able to more accurately estimate the function described by the simulator, compared to the MAF metamodel. We find that posterior distributions estimated from the DGPR metamodel is more accurate compared to the ones estimated from the MAF metamodel, but takes longer to compute. We estimate the posterior distributions separately for DGPR metamodels trained on the population spiking activities and the LFP, for a variety of different simulation outputs, and find that some parameters are significantly more constrained by observing the LFP compared to the population spiking activities, while for others both signals are roughly equally informative. We also investigate the pairwise correlations of the posterior distributions over the domain on which the metamodels were trained.

## 2 Methods

### 2.1 Simulations

The network considered in this study consists of two populations, one with excitatory neurons and one with inhibitory neurons, of size $N_E$ and $N_I$ respectively. The neurons are modelled as leaky integrate-and-fire neurons, connected with alpha-shaped current-based synapses described by Eq 2 in Table 1. The sub-threshold dynamics are described by a first-order differential equation given by Eq 1 in Table 1. When its membrane potential reaches a firing threshold $V_{\text{thr}}$, the neuron fires and and the synapses on all its postsynaptic neurons are activated after a time $t_d$. After firing, the membrane potential of a neuron is clamped to a voltage $V_{\text{reset}}$ for a refractory period $t_{\text{ref}}$. Each neuron receives synapses from a fraction $\epsilon$ of all other local neurons in the network. Post-synaptic currents from excitatory synapses have a peak amplitude $J$, and post-synaptic currents from inhibitory synapses have a peak amplitude of $-gJ$. Both types of synapses share a synaptic time constant of $\tau_{\text{syn}}$. Additionally, each neuron receives independent excitatory input from an external population modelled as a Poisson process with fixed rate $v_{\text{ext}}$. This rate is determined by the parameter $\eta = \frac{v_{\text{ext}}}{v_{\text{thr}}}$, where $v_{\text{thr}} = V_{\text{thr}}C_{\text{m}}/(J\tau_{\text{m}}e\tau_{\text{syn}})$, the minimum constant input rate that will drive the neuron to its threshold. Since the synaptic strength and synaptic time constant both contribute multiplicatively to the total charge transfer during a synaptic event, we opted to parameterize the synaptic strength in terms of the total charge transfer during a synaptic event $Q_{\text{s}} = \tau_{\text{syn}}J$. A complete description of the point-network model is given in Table 1.

Ten model parameters were uniformly randomly sampled, using Latin Hypercube sampling [27], which is a stratified sampling strategy suitable for sampling in high-dimensional systems. A complete list of model parameters and their values and ranges are given in Table 2. 10000 simulations were run using the NEST simulator [28, 29], comprising both the training data set and the test data set. The regions of the parameter space giving strongly synchronous firing, discussed in [5, 30], were avoided by avoiding unbalanced regimes caused by low inhibition, and high external input ($\eta > 3.5$). Additionally, a small number of simulations, which were found to be strongly synchronous were removed, making up roughly 3% of the total simulations. These were found by the following ad hoc criteria: more than 150 time bins of 1 ms in which more than 10% of the neurons fire, as well as more than 500 time bins in which fewer than 0.25% of the neurons fire. See Section 3.1 and Section 4 for further discussion. For 100 of the simulation outputs in the test data set, an additional 50 simulations were run with parameters sampled from the estimated posterior distributions (see Sections 2.4 and 3.4). For each simulation, the LFP was computed as described in the next section.

**Table 1. Description of point-neuron network following the guidelines of [33].**

| A | Model summary |
|---|---|
| **Populations** | One excitatory, one inhibitory |
| **Network model** | Fixed in-degree, random convergent connections |
| **Neuron model** | Local populations: leaky integrate-and-fire, external: Poisson generator |
| **Synapse model** | Current-based alpha-shaped, fixed strength for each population |
| **B** | **Populations** |
| **Names** | Excitatory: E |
| | Inhibitory: I |
| **C** | **Network model** |
| **Connectivity** | Fixed number of incoming connections $C_E = \epsilon N_E$ from excitatory population and $C_I = \epsilon N_I$ from inhibitory population |
| **Input** | Poissonian synaptic input with fixed rate $\nu_{ext}$ for each neuron |
| **D** | **Neuron model** |
| **Type** | Leaky integrate-and-fire neuron |
| **Description** | Dynamics of membrane potential $V_i(t)$ (neuron $i \in [1, N]$): <br> - Spike emission at times $t_i^i$ with $V_i(t_i^i) \geq V_{thr}$ <br> - Subthreshold dynamics: <br> $$\tau_m \frac{dV_i(t)}{dt} = -V_i(t) + R_m I_i(t) \quad \forall l : t \notin (t_l^i, t_l^i + t_{ref}] \quad (1)$$ <br> where $\tau_m$ is the membrane time constant, $V$ the membrane potential, $R_m$ the membrane resistance, and $I$ the synaptic inputs. <br> - Reset + refractoriness: $V_i(t) = V_{reset} \, \forall l : t \in (t_l^i, t_l^i + t_{ref}]$ <br><br> Exact integration with temporal resolution $dt$ <br> Uniform distribution of membrane potentials $V_i \in [V_{reset}, V_{thr}]$ at $t = 0$ |
| **E** | **Synapse model** |
| **Type** | Alpha-shaped postsynaptic current |
| **Description** | $$R_m I_i(t) = \tau_m \sum_j \frac{J_{ij}}{\tau_s} \sum_l H(t - t_l^j - t_d) t e^{1 - (t - t_l^j - t_d)/\tau_{syn}}, \quad (2)$$ <br> where the first sum is over all the presynaptic neurons $j$, including the external ones, and the second sum is over the spike times of those neurons. $t_l^j$ is the $l$th spike of presynaptic neuron $j$, and $t_d$ is the synaptic delay. $H$ denotes the Heaviside step function. <br> $$J_{ij} = \begin{cases} J, & j \in \{E, E_{ext}\} \\ -gJ, & j \in \{I\} \end{cases}$$ <br> Multapses and autapses are allowed. |

The population spiking activities were saved at a resolution of 1 ms as the sum of all spike trains in the populations. The power spectrum, defined as $[\psi](f)^*[\psi](f)/T$ for some signal $\psi(t)$, where $[\psi](f)$ denotes the Fourier transform of $\psi$, and $T$ is the window length, was estimated using Welch's method [31] for both the population spiking activities and the LFP. The implementation in the Python SciPy [32] package (`scipy.signal.welch`) was used, with a Hann window of length 256 and an overlap of 128. The metamodels are trained on the base 10 logarithm of these power spectra.

## 2.2 Forward-model predictions of LFPs

In order to compute local field potentials (LFPs) from the point-neuron network, we utilized the hybrid LFP scheme [34] (github.com/INM-6/hybridLFPy), which allows for the decoupling of the simulation of spiking dynamics (here computed using point neurons) and

**Table 2. Point-neuron network parameters.**

| Symbol | Description | Value |
|---|---|---|
| **Point-neuron parameters** | | |
| $\eta$ | relative amount of external input | [1.0, 3.5] |
| $g$ | relative strength of inhibitory synapses | [4.5, 8.0] |
| $Q_s$ | total synaptic charge transfer | [25 − 100] fC |
| $\tau_m$ | membrane time constant | [15–30] ms |
| $C_m$ | membrane capacitance | [100 − 300] pF |
| $t_d$ | synaptic delay period | [0.1 − 3.0] ms |
| $t_{ref}$ | absolute refractory period | [0.1 − 4.0] ms |
| $\tau_{syn}$ | synaptic time constant | [1.0 − 8.0] ms |
| $V_{thr}$ | firing threshold | [15 − 25] mV |
| $V_{reset}$ | reset membrane potential | [0 − 10] mV |
| $E_L$ | passive leak reversal potential | 0 mV |
| $N_E$ | number of excitatory neurons | 8000 |
| $N_I$ | number of inhibitory neurons | 2000 |
| $\epsilon$ | connection probability | 0.1 |
| $C_E$ | number of incoming excitatory synapses | 800 |
| $C_I$ | number of incoming inhibitory synapses | 200 |
| **Simulation parameters** | | |
| Training and test data | | |
| $T_{sim}$ | simulation duration | 10500 ms |
| $T_{transient}$ | start-up transient duration | 500 ms |
| $dt$ | time resolution | 0.1 ms |

predictions of extracellularly recorded LFPs. The latter part relies on reconstructed cell morphologies and multi-compartment modelling in combination with an electrostatic forward model. A complete description of the scheme (including the biophysics-based forward model) can be found in [34], and a concise description of the multi-compartment neuron model is given in Table 3. The parameters used are given in Table 4.

Current-based synapses and morphologies with passive membranes in the multi-compartment neuron models give a linear relationship between any presynaptic spike event and contributions to the LFP resulting from evoked currents in all postsynaptic multi-compartment neurons. Thus the LFP contribution $\phi_Y^j(\mathbf{r}, t)$ at position $\mathbf{r}$ from a single presynaptic point-neuron $j$ in population $Y$ can, in general, be calculated by the convolution of its spike train $v_Y^j(t) \equiv \sum_k \delta(t - t_j^k)$ with a unique kernel $H_Y^j(\mathbf{r}, \tau)$ as $\phi_Y^j(\mathbf{r}, t) = (v_Y^j * H_Y^j)(\mathbf{r}, t)$. This kernel encompasses effects of the postsynaptic neuron morphologies and biophysics, the electrostatic forward model, the synaptic connectivity pattern, conduction delay and post-synaptic currents. This kernel can be further decomposed as a convolution between the LFP response of a unit current for a single time step, and the post-synaptic current. The resulting LFP due to spikes in a presynaptic population $Y$ is then given by

$$\phi_Y(\mathbf{r}, t) = \sum_{j \in Y} (v_Y^j * h_Y^j * I_Y^j)(\mathbf{r}, t) , \tag{9}$$

where $I_Y^j(t)$ is the post-synaptic current given by Eq 7 in Table 3, and $(h_Y^j * I_Y^j)(\mathbf{r}, t) = H_Y^j(\mathbf{r}, t)$.

**Table 3. Description of multi-compartment neuron populations.**

| A | Model summary |
|---|---|
| **Populations** | Local excitatory and inhibitory populations |
| **Neuron model** | Multi-compartment neurons with passive cable formalism |
| **Synapse model** | Current-based alpha-function shaped, fixed strength for each population |
| **Topology** | Cylinder of 1 mm$^2$ cross-section with somas of both populations positioned in single layer of thickness 0.1 mm. |
| **B** | **Neuron models** |
| **Type** | Reconstructed multi-compartment morphologies with passive electrical properties |
| **Description** | For each neuron, the membrane potential $V_n$ of compartment $n$ connected to $m$ other compartments $k$, with a surface area $a_n$, length $l_n$ and diameter $d_n$ is given by: $$\sum_{k=1}^{m} g_{akn}(V_k - V_n) = C_{mn}\frac{dV_n}{dt} + I_{mn} \quad (3)$$ $$C_{mn} = c_m a_n \quad (4)$$ $$g_{akn} = \pi(d_n^2 + d_k^2)/(4r_a(l_n + l_k)) \quad (5)$$ $$I_{mn} = g_{Ln}(V_n - E_L) + \sum_j I_{jn}, \quad (6)$$ where for compartment $n$, $C_{mn}$ is the membrane capacitance, $g_{akn}$ the axial conductance from compartment $k$, $I_{mn}$ the membrane current, $g_{Ln}$ the membrane leak conductance, $E_L$ the extracellular reversal potential, and $I_{jn}$ the synaptic current from presynaptic neuron $j$. |
| **C** | **Synapse model** |
| **Synapse type** | $\alpha$-function shaped postsynaptic current |
| **Description** | $$I(t) = H(t - t_a)JCte^{1-t/\tau_{syn}} \quad (7)$$ $$H(t) = 0 \text{ for } t \leq 0, \text{ otherwise } 1. \quad (8)$$ Here, $t_a$ is the activation time of the synapse, $J$ the synaptic strength, and $\tau_{syn}$ is the synaptic time constant. $C$ is a constant chosen so that $JC\int_0^\infty te^{1-t/\tau_{syn}}dt = C_m J$, assuring that the same total charge is transferred as in the $\delta$-function synapse in the point-neuron network. |
| **D** | **Topology** |
| **Type** | Cylinder with radius $1/\sqrt{\pi}$ mm and height 0.5 mm containing two vertical sections |
| **Description** | - Cylinder extends from $z = -500\,\mu$m to $z = 0$<br>- All somas are randomly placed with a uniform distribution within the boundaries $r \leq 564\,\mu$m and $-450\,\mu$m $\leq z \leq -350\,\mu$m<br>- Two regions separated by the plane $z = -300\,\mu$m<br>- Synapses on inhibitory neurons are placed in lower region<br>- Inhibitory synapses on excitatory neurons are placed in lower region<br>- Excitatory synapses on excitatory neurons are split equally between regions |

**Table 4. Multi-compartment neuron parameters.**

| Multi-compartment neuron parameters | | |
|---|---|---|
| **Symbol** | **Description** | **Value** |
| $\tau_m$ | membrane time constant | [15, 30] ms |
| $c_m$ | membrane capacitance | [0.6, 1.4] μF/cm$^2$ |
| $r_m$ | membrane resistivity | $\tau_m/c_m$ |
| $R_a$ | axial resistivity | 150 Ωcm |
| $\tau_{syn}$ | synaptic time constant | [1.0, 8.0] ms |
| $E_L$ | passive leak reversal potential | 0 mV |
| $V_{init}$ | membrane potentials at $t = 0$ ms | 0 mV |
| $\sigma_e$ | extracellular conductivity | 0.3 Sm$^{-1}$ |

As shown in [30, 34], a good approximation can be made by convolving the population firing rates $v_Y(t) \equiv \sum_{j \in Y} v_Y^j(t)$ with averaged kernels $\overline{H}_Y(\mathbf{r}, \tau) \equiv 1/N_Y \sum_{j \in Y} H_Y^j(\mathbf{r}, \tau)$, that is,

$$\phi_Y(\mathbf{r}, t) = (v_Y * \overline{H}_Y)(\mathbf{r}, t) . \tag{10}$$

As in [30, 34] these averaged kernels $\overline{H}_Y(\mathbf{r}, \tau)$ were computed using the full hybrid-scheme. This was done by computing the LFP resulting from a fully synchronous activation of all the outgoing synapses from all neurons in the presynaptic population. Thus for the computation of the LFP kernel, we have $v_Y^j(t) \equiv \delta(t - t_Y)$ where $t_Y$ is the timing of the synchronous event in population $Y$.

After the single time-step kernel has been computed, the full kernel can easily be computed for any post-synaptic current, and hence synaptic parameters. In order to account for the membrane parameters $\tau_m$ and $c_m$, where $c_m$ is the per-area capacitance, we compute the kernel for a single time step on a grid of different values of $\tau_m$ and $C_m$, and adjust this kernel by linear interpolation. We compute the kernels at the values [0.4, 0.6, 0.8, 1.0, 1.2] for $c_m$ and [15, 18, 21, 24, 27, 30] for $\tau_m$. Since the membrane capacitance for morphological neurons is dependent on the area of the neuron, we choose a baseline value of $c_m = 1.0$, and adjust it by the same proportion as $C_m$ for the point-neuron network. The synaptic parameters can subsequently be accounted for in a separate step.

## 2.3 Gaussian process regression

Here we give a brief overview of Gaussian process regression models, their training procedures and the extension to deep models.

Gaussian process regression models are probabilistic non-parametric models that provide distributions over arbitrary functions, and are trained using supervised learning. A Gaussian process (GP) is a collection of random variables such that any finite number of them follow a joint Gaussian distribution, and a GP is completely specified by a *mean function $m(\boldsymbol{\theta})$* and a *covariance function $k(\boldsymbol{\theta}, \boldsymbol{\theta})$* [18]. Consider a data set consisting of $n$ inputs of dimension $d$ contained in the matrix $\Theta$, and corresponding outputs $\mathbf{x}$, which are assumed to be one-dimensional for notational simplicity in this section. A Gaussian process regression model assumes a distribution over an underlying latent function $f(\boldsymbol{\theta})$ which is modelled as a Gaussian process. Evaluating this distribution at the set of input points $\Theta$ gives a joint Gaussian distribution $f(\Theta) \sim \mathcal{N}(\mathbf{0}, k(\Theta, \Theta))$, where the covariance function $k(\Theta, \Theta)$ must return a positive semi-definite covariance matrix between all the data points contained in $\Theta$. The mean function is usually assumed to be 0 without loss of expressiveness. The covariance function encodes the properties of the function, and there are many different covariance functions available with different properties. In the present work, we use the Matérn 5/2 kernel

$$C_{5/2}(\boldsymbol{\theta}, \boldsymbol{\theta}') = \sigma^2 \left( 1 + 5d + \frac{5}{3} d^2 \right) \exp(-5d), \quad d = \sqrt{5 \sum_i \left( \frac{\theta_i - \theta_i'}{l_i} \right)^2}, \tag{11}$$

where $\mathbf{l}$ and $\sigma$ are learnable parameters.

The model of the observed outputs $\mathbf{x}$ is given by the underlying latent function $\mathbf{f}(\Theta)$ corrupted by some noise $\epsilon \sim \mathcal{N}(\mathbf{0}, \sigma_n^2 I)$ such that $p(\mathbf{x}|\mathbf{f}) = \mathcal{N}(\mathbf{f}, \sigma_n^2 I)$. The marginal likelihood of the data is obtained as

$$p(\mathbf{x}; \Theta) = \int p(\mathbf{x}|\mathbf{f}; \Theta) p(\mathbf{f}; \Theta) d\mathbf{f} = \mathcal{N}(\mathbf{0}, (K(\Theta, \Theta) + \sigma_n I)^{-1}), \tag{12}$$

which can be found analytically in the case of a Gaussian likelihood $p(\mathbf{x}|\mathbf{f}, \Theta)$, and is used to optimize the model parameters.

Given a new set of inputs $\Theta_*$ on which we would like to evaluate the model, the GP specifies a joint distribution over $\mathbf{x}$ and $\mathbf{x}_*$, and the predictions are made as the distribution over $\mathbf{x}_*$ conditioned on the training data $\mathbf{x}$ [18]

$$p(\mathbf{x}_*|\mathbf{x}; \Theta_*, \Theta) = \int p(\mathbf{x}_*|\mathbf{f}_*; \Theta_*)p(\mathbf{f}_*|\mathbf{f}; \Theta, \Theta_*)p(\mathbf{f}|\mathbf{x}; \Theta)d\mathbf{f}d\mathbf{f}_* \tag{13}$$

$$= \mathcal{N}(\mathbf{x}_*|K_{n*}K_{nn}^{-1}\mathbf{x}, \quad K_{**} - K_{f*}^T(K_{nn} + \sigma_n^2 I)^{-1}K_{f*} + \sigma_n^2 I), \tag{14}$$

where we use the shorthand notation $K_{nn} = k(\Theta, \Theta)$, $K_{n*} = k(\Theta, \Theta_*)$ and $K_{**} = k(\Theta_*, \Theta_*)$. The semicolon indicates the input points corresponding to the distributions, on which the covariance function is evaluated. We can immediately see that the mean value of the distribution at the predicted locations $\Theta_*$ are linear combinations of the training values $\mathbf{x}$.

Deep, or composite, GP models are constructed by iteratively defining GPs with the previous GP as inputs. A joint prior distribution over all layers is given by

$$p(\mathbf{x}, \{\mathbf{f}^l, \}_{l=1}^L) = \prod_{i=1}^N p(x_i|f_i^L)\prod_{l=1}^L p(\mathbf{f}^l; \mathbf{f}^{l-1}) \ , \tag{15}$$

where $\mathbf{f}^0 = \boldsymbol{\theta}$. As in standard GP models, we would like to marginalize out the latent functions in all layers for optimization. For predictions, we would like to condition the latent functions on the training data and then marginalize them. This is obviously not possible analytically, and approximations have to be used. We do not go in to details, but refer the readers to [35] for an excellent tutorial.

For multi-dimensional outputs, one can either assume the outputs to be independent, or correlated. In the former case, a separate GP is indepedently trained for each output dimension. For the latter case, one simple way of introducing correlations between output channels is to make a linear combination of independent GPs. If the number of independent GPs is lower than the number of output dimensions, a low-dimensional representation of the output is achieved, with a linear mapping between the latent space and the output. The model can be written as

$$\mathbf{g}(\boldsymbol{\theta}) = \{g_p(\boldsymbol{\theta})\}_{p=1}^P, \qquad \mathbf{f}(\boldsymbol{\theta}) = W\mathbf{g}(\boldsymbol{\theta}), \tag{16}$$

where $g_p(\boldsymbol{\theta})$ forms a set of independent GPs, and $W$ is an $n \times p$ matrix used to linearly construct the output $\mathbf{f}(\boldsymbol{\theta})$ [36].

This can be viewed as an input-dependent linear latent space model, where we assume a linear mapping between a latent space and the output, but the coefficients of this mapping can change in a non-linear fashion. If we choose $P$ to be a small number, a low-rank representation of the output is achieved.

We used a DGPR model based on the framework by [23]. We use 2 layers, and do a parameter scan in order to find the number of GPs to use in each layer. The final output is constructed by a linear transformation from the GPs in the second layer as given by Eq 16.

## 2.4 Posterior distribution over parameters

The metamodels provide a distribution $p(\mathbf{x}|\boldsymbol{\theta})$, which is an approximation to the distribution over the outputs given by the simulations, where $\mathbf{x}$ is the simulation output, and $\boldsymbol{\theta}$ is the simulation inputs. Introducing a prior distribution over the inputs $p(\boldsymbol{\theta})$, which we take to be uniform over the ranges of input values used to train the metamodels, the unnormalized posterior distribution over parameters given a specific observed model output $\mathbf{x}_0$ is given by $p(\boldsymbol{\theta}|\mathbf{x}_0) \propto p$

$(\mathbf{x}_0|\boldsymbol{\theta})p(\boldsymbol{\theta})$. We sampled from this distribution using an adaptive Metropolis algorithm [37]. The adaptive Metropolis algorithm is a random walk Markov Chain Monte Carlo algorithm, where a multivariate Gaussian proposal distribution is used, and where its covariance matrix is iteratively estimated by the sample covariance. We only estimated the covariance during the burn-in period, and kept it constant during sampling. Additionally, we used a separate scaling factor for the covariance matrix which was tuned during the burn-in period to give an acceptance ratio of around 0.3. Five chains were run in parallel, and run for a total of 40000 evaluations of the proposals, where the samples from the first 12000 proposals were discarded and considered the burn-in period. For each example in the test data set, the posterior distribution was sampled separately for both the metamodel of the population spiking activities and the metamodel for the LFP. The Gelman-Rubin statistic [38] was computed independently for each parameter, for each distribution, in order to assess the convergence properties of the chains.

## 2.5 Technical details

**2.5.1 Reproducibility.** The simulations were run using Python v3.10.5. All point-network simulations were run with the NEST simulator v3.3 [28, 29]. The forward-modeling of the LFP was done using hybridLFPy v0.2 [34] with LFPy v2.3 [39]. The DGPR metamodels were trained using GPyTorch v1.9.1 [40] with PyTorch v1.13.1. The MAF metamodels were trained using SBI v0.21.0 [41]. The metamodels were trained on NVIDIA Quadro RTX 8000 GPUs. All simulation and analysis code for this study is available on Github: https://github.com/janskaar/Skaar_bioRxiv_506616v1

# 3 Results

The aim of this study was to apply metamodelling techniques to simulations of neuronal networks. Specifically, we applied Deep Gaussian process regression (DGPR) and masked autoregressive flow (MAF) models to approximate the likelihood of the network simulator model. In all cases, we consider the output of the simulator to be the power spectrum of either the summed population spiking activity, or the local field potential generated by the network. We compare the accuracies of the forward predictions of all three models on the population spiking activities. The DGPR metamodel is trained on both the LFP and the population spiking activities in order to evaluate the differences in information in the two signals. For the DGPR and MAF metamodels, we also estimate the posterior distribution for 100 examples in the test data set, and evaluate the accuracy of these based on the posterior predictive distribution.

We start by showing some example simulation outputs, and discuss the accuracy of the forward predictions of the metamodels. We then evaluate the number of training samples needed for the metamodels to obtain high prediction accuracy, before we discuss the inverse modelling and the posterior distributions. A HCPLSR metamodel was also trained on the same task. We refer to S3 Supplementary Section. for the results from that model.

## 3.1 Simulations

A total of 10000 simulations were run, with parameters sampled as described in Section 2.1. A network consisting of one excitatory and one inhibitory population can produce a variety of activity states, characterized by whether the neurons fire in *synchrony*, and how *regularly* they fire, that is, whether the inter-spike intervals for individual neurons have a wide or narrow distribution. The behaviour of the present two-population network was analyzed in the framework of mean-field theory in [5] for integrate-and-fire neurons where the synaptic currents were modelled as delta functions. Brunel [5] found that the network spiking activity could be

described by distinct regimes, classified as synchronous regular (SR), asynchronous irregular (AI) and synchronous irregular (SI, fast and slow), depending on which specific parameter values were used, and arise by different mechanisms. As described in Section 2.1, we avoided activity states characterized by strongly synchronous behaviour, which produces sharp peaks in the power spectrum. Fig 2 shows three examples of different states of network activity. Column A shows the synchronous regular regime, characterized by runaway excitation, where the synaptic strengths of inhibitory neurons are not strong enough to overcome the excitation, and the neurons fire regularly and synchronously with very high rates. Column B shows an example of a synchronous irregular state, produced in some cases when the network receives high amounts of external input and the inhibitory synaptic strengths are high. Column C shows asynchronous irregular activity. This is the regime in which most of the simulations used for training and validation were, although there is of course much variation within this regime. Columns A and B show the type of activity we avoided, with sharply peaked spectra.

We will not attempt to give a full description of all possible network states or the effects of all different parameters in this paper, but show these examples to make clear the states we have actively avoided, as described in Section 2.1.

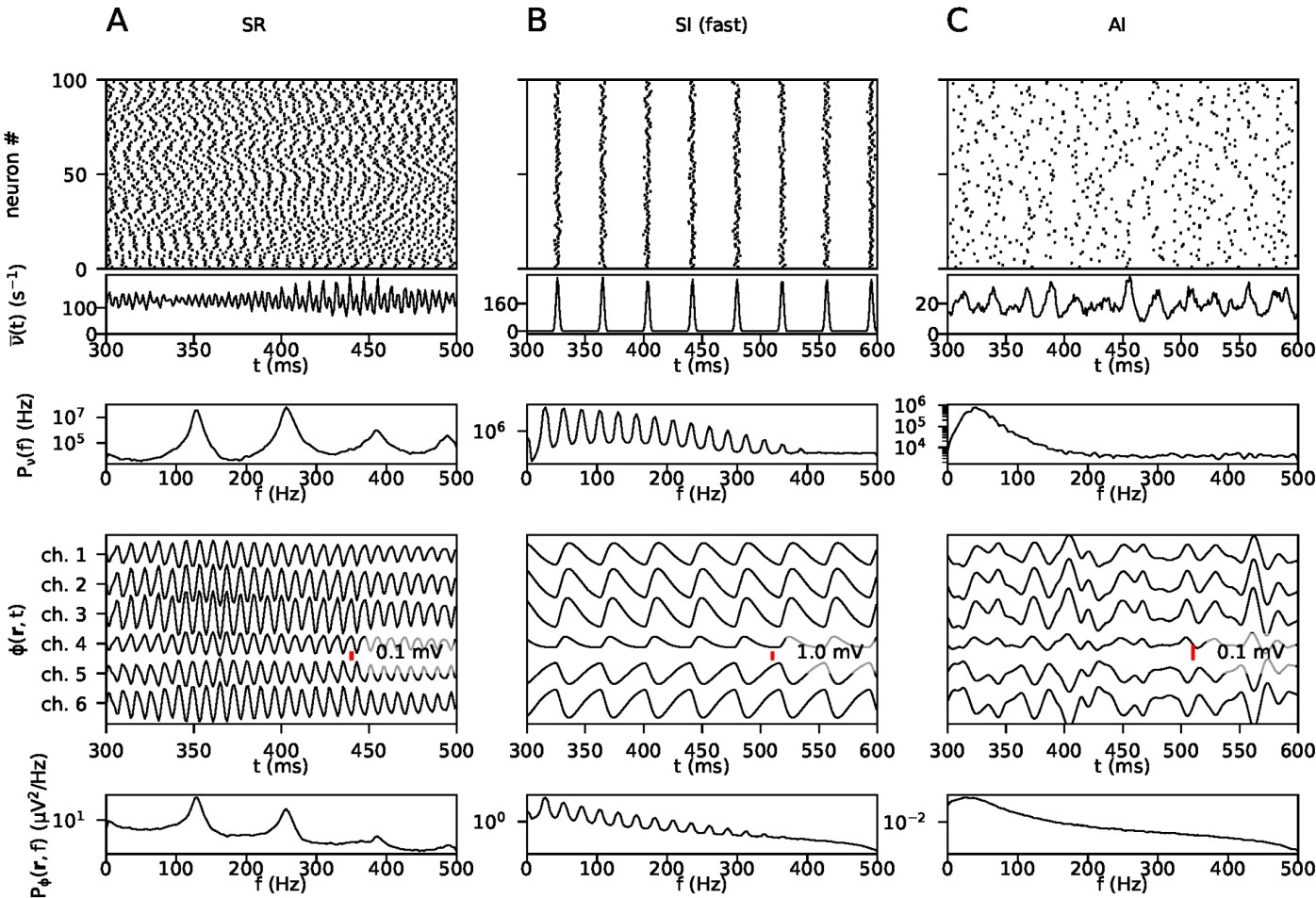

**Fig 2. Example activity states.** Each column shows, from top to bottom, a raster plot, population firing rates, the power spectrum of the population averaged firing rates, the LFP, and the power spectrum of the LFP. The raster plot and firing rates are taken from the excitatory population. A: Synchronous regular activity. B: Synchronous irregular activity. C: Asynchronous irregular activity.

## 3.2 Forward predictions

The metamodels were tested on a data set containing 1681 examples, jointly sampled with the training data using Latin Hypercube sampling. Models were trained on data sets ranging from 100 to 8000 examples in order to examine the density of samples needed to adequately represent the simulation output space. For all other experiments, the metamodels used were trained on the full 8000 example simulations. A grid search of hyperparameters was run for each metamodel in order to find a suitable model for the problem at hand, which can be found in S1 Supplementary Section. The metamodel obtained from the hyperparameter search was used for all subsequent experiments, and where we trained new metamodels, the hyperparmeters were the kept the same. Fig 3A and 3B show examples of simulated outputs and corresponding outputs from the metamodels for two different sets of parameter values, for the population spiking activities and LFPs respectively. Only the outputs for the excitatory population spiking activity and the topmost LFP channel are shown. 10 simulations run with different seeds are shown in black lines, and the colored lines show the output given by the metamodels. The shaded area shows two standard deviations above and below the mean for the DGPR and MAF metamodels. Visually inspecting the top example in panel A, both metamodels accurately capture the mean of the simulations for all frequencies, but slightly overestimate the variance, particularly for the lower frequencies. In the bottom example, there is a larger discrepancy between the MAF metamodel and the simulator for the middle frequencies, while the DGPR metamodel is accurately able to capture both the mean and variance. Note that the covariance between frequency channels is not considered at all in this figure. In panel B, the LFP for the same simulations and the DGPR prediction are shown. Generally, the LFPs are smoother and the variance is lower compared to the population spiking activity. Fig 3C and 3D show the distribution over maximum (over frequencies) absolute errors on the test data set, where the expectation of the metamodel is used as the prediction value. We show the maximum error over frequencies since the large number of output channels make the mean small even in the presence of a few large deviations. Moreover, as the simulation output itself is random, we are mainly interested in finding any larger deviations, rather than small deviations, which are expected to occur. The expectation of the distribution given by the DGPR metamodel more closely matches the simulated values compared to the MAF metamodel. Fig 3E and 3F show the mean standard deviation of the metamodel distributions evaluated at the test data parameters. The DGPR metamodel produces narrower distributions over the simulator output compared to the MAF model.

In terms of computational cost, evaluating the log probability of 50 example outputs takes around 4 seconds with the DGPR metamodel, and 0.014 seconds with the MAF metamodel. Generating 50 samples from the metamodels takes 1.6 seconds for the MAF metamodel, and 0.07 seconds with the DGPR metamodel. By contrast, running a single simulation took on average 4.4 CPU-minutes.

The training time for the metamodels depends on the model parameters and learning rate, as well as the random seed. A full epoch of the 8000 training examples takes around 4.4 seconds for the DGPR metamodel, and 1.4 seconds for the MAF model. Generally, the training converged after a few hundred epochs for both models, giving a training time (not counting the periodic train/test evaluation) on the scale of an hour for the DGPR model and around a third of that for the MAF model.

## 3.3 Number of training samples needed to represent the model

The DGPR metamodel was trained on data sets ranging from 100 to 8000 examples of the population spiking activities, and the performances on the same test data set were evaluated. Fig 4

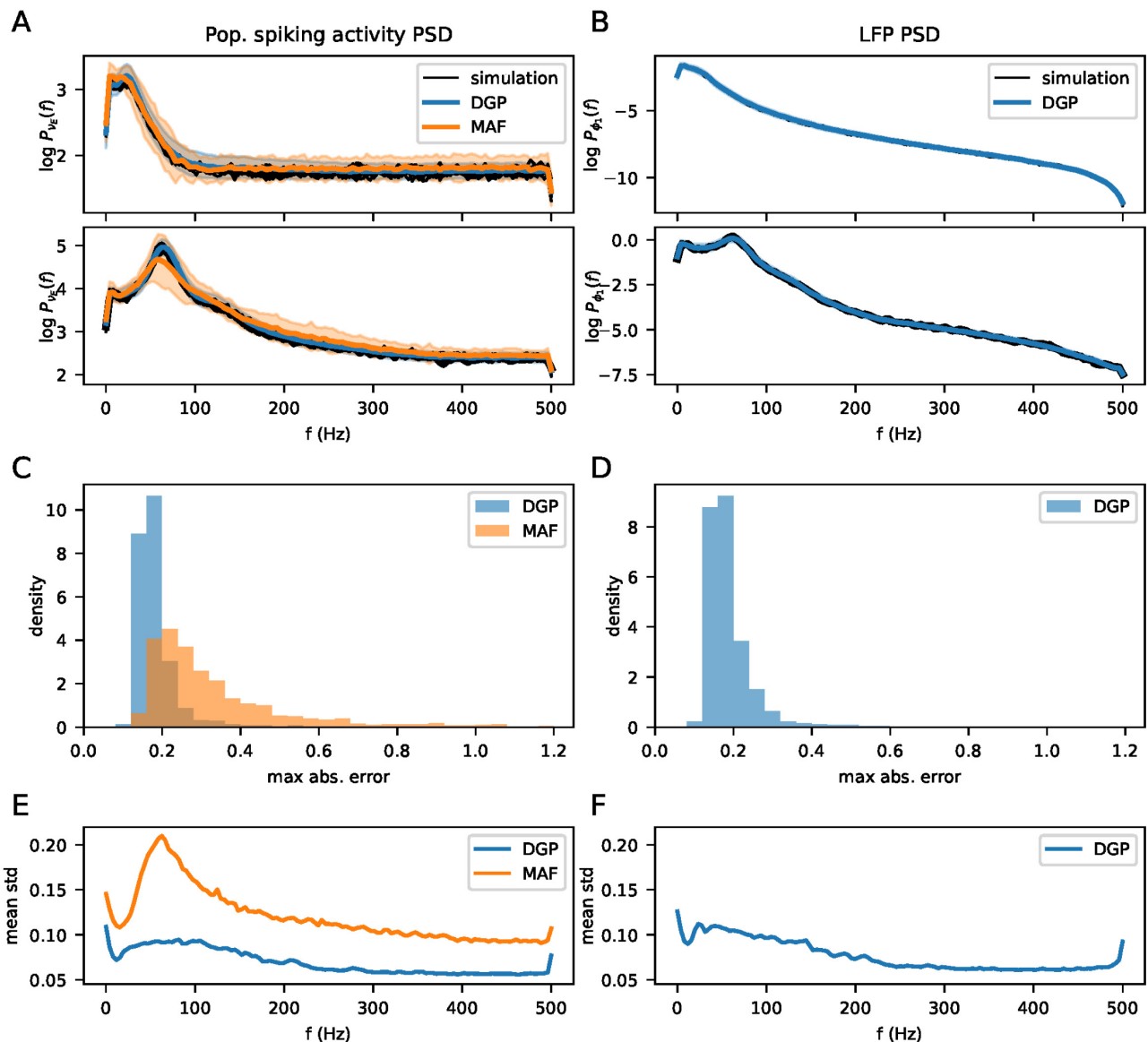

**Fig 3.** A: For two different parameter sets, the power spectrum of the population spiking activity from 10 simulations (black lines), and metamodels (blue and orange lines) are shown. The shaded area shows two standard deviations of the distribution given by the metamodels. B: Same as A, but for the LFP. Only the DGPR metamodel was trained on the LFP. C: Distribution over maximum (over frequencies) absolute errors for both metamodels, on the power spectrum of the population spiking activity. D: Same as for C, but for the LFP. E: Mean standard deviation of the metamodel outputs, evaluated at the parameters in the test data set. F: Same as E, but for the LFP.

shows the maximum (over frequencies) absolute prediction errors, averaged over the test data set for each trained metamodel. As the number of training samples increases, the prediction errors on the test data set decreases, as the metamodel starts to generalize better. The prediction errors on the training data set increases, as it becomes more heterogeneous and the model is less able to overfit to it. After a sharp decrease in errors as the number increases to a few hundreds, the benefit of increasing the number of training samples decreases significantly after on the order of 1000 is reached, indicating that the parameter space is sufficiently densely sampled.

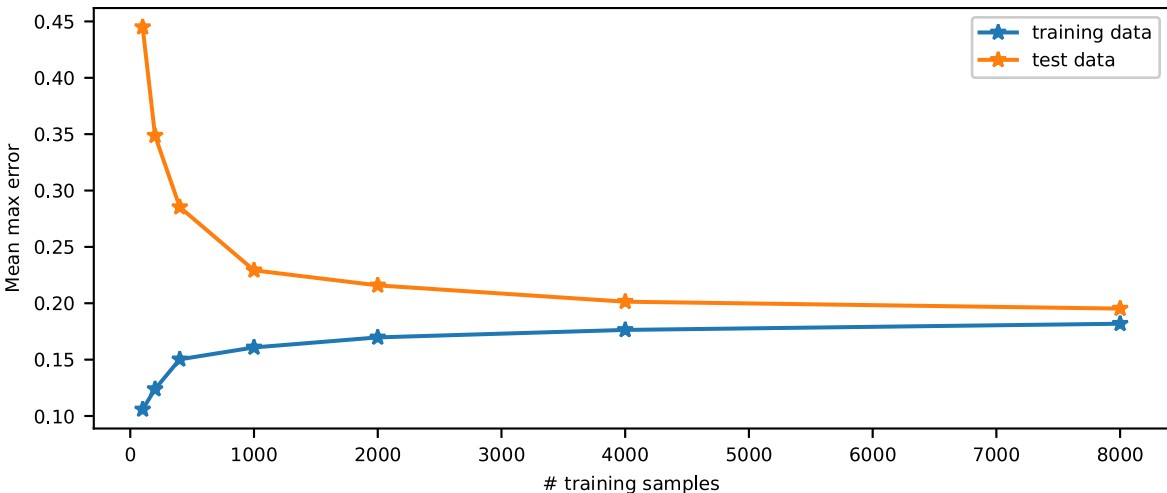

**Fig 4.** Mean (over test data set and training data set) of the maximum (over frequencies) errors, in absolute scale, as a function of the number of examples used to train the metamodels on the population spiking activities.

## 3.4 Inverse modelling

Using the conditional distribution over simulation outputs given parameter values, estimated by the metamodels, posterior distributions for 100 simulations in the test data set were estimated. Separate posterior distributions for both metamodels conditioned on the population spiking activities, and for the DGPR metamodel conditioned on the LFP, were estimated using an adaptive Metropolis algorithm as described in Section 2.4.

The Gelman-Rubin statistic was computed for all the distributions, and was below 1.1 for over 90% of the posterior distributions for the population spiking activities, and over 95% of the posterior distributions for the LFP, indicating that the chains converged for most of the distributions. In the computation of average values over the test data set, all distributions were used, regardless of the value of the Gelman-Rubin statistic. For each of the estimated posterior distributions of both the population spiking activities and the LFP, 50 additional simulations were run with parameters drawn from the posterior distributions in order to evaluate the accuracy of the posterior distribution.

Fig 5A shows an example of a posterior distribution generated by the outputs of one simulation in the test data set. The 1D and 2D posterior marginal distributions of a subset of the parameters are shown. On the left-hand side, the posterior distribution generated by the population spiking activities are shown, and the corresponding posterior distribution from the LFP is shown on the right-hand side. The red bars and dots show the values of the parameters used in the simulation from which the posterior is generated. Qualitatively, the distributions from both the population spiking activities and the LFP appear similar for the first three parameters shown.

For this particular example, the 1D marginal distributions of both posteriors are wide for the membrane time constant $\tau_\mathrm{m}$, quite narrow for $g$, and somewhere in between for $\eta$. Wide marginal distributions entail that, lacking information about all other parameters, the value of the parameter cannot be determined with certainty, while narrow marginal distributions entail that the parameter can be accurately determined even without knowledge of other parameters. There is a positive correlation between $\eta$ and $\tau_\mathrm{m}$ in the posterior distribution, i.e. if the value of $\eta$ is increased the value of $\tau_\mathrm{m}$ must also be increased for the model to produce similar outputs.

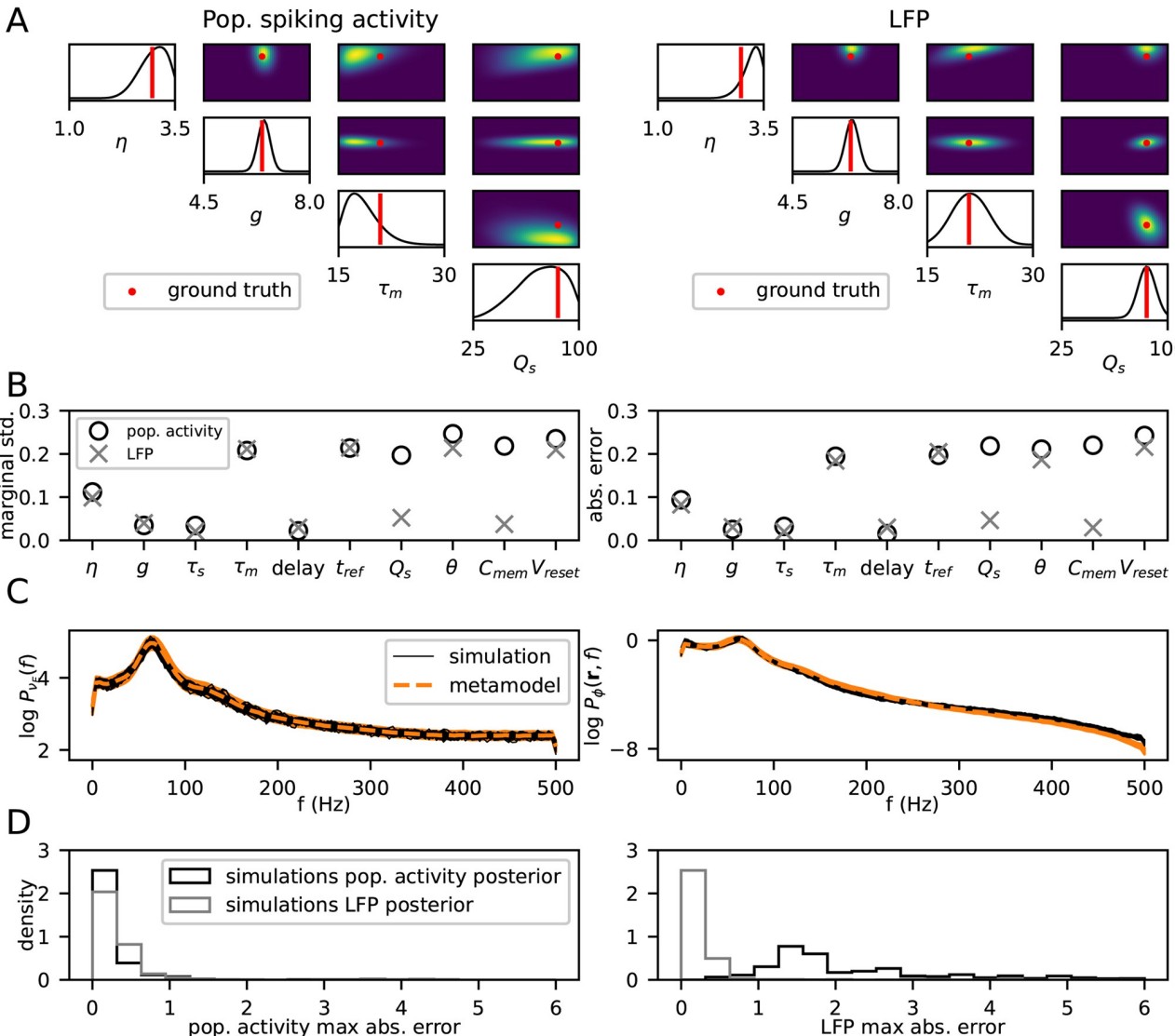

**Fig 5.** In A, C and D, the left-hand and right-hand side show equivalent plots for the population spiking activities and LFPs respectively. The DGPR metamodel is used in both cases. A: 1D and 2D marginal posterior distributions over a subset of the parameters, for an example in the test data set. The red dots and bars show the parameter values corresponding to the simulation output the posterior distribution was computed for. B: Left: mean (across test data set) of standard deviation of the 1D marginal distributions for each parameter. Right: Absolute error of the parameter predictions based on the expectation. C: Metamodel predictions for the power spectra of excitatory population spiking activity (left) and the uppermost channel of the LFP (right) in orange. Black lines show 50 simulation outputs run with parameters drawn from the posterior distribution. D: Distribution of maximum distance between the ground-truth simulation output from which the posterior distribution was computed, and the simulation output from the posterior distributions (black lines in C). Left plot shows the errors from the population spiking activities, right plot shows the errors from the LFP. The black lines show the simulations from the posterior conditioned on the population spiking activities, the gray lines show the simulations from the posterior conditioned on the LFP.

Equivalently, if you know the specific value of either $\eta$ or $\tau_\mathrm{m}$, the uncertainty in the other parameter would decrease. Note that because we only see 1D and 2D marginals, there may be higher-order interactions present that cannot be easily visualized. For the synaptic strengths, the distribution is much wider for the model trained on the population spiking activities compared to the model trained on the LFP.

The left-hand side of Fig 5B shows the mean (over all test examples) standard deviations of the 1D marginal distributions. Generally, the parameters related to the synaptic inputs have the narrowest marginal distributions, while the neuron parameters have much wider marginal distributions, indicating that changes to neuron parameters can largely be compensated for by adapting other parameters, while changes to synaptic and input parameters have a larger effect that cannot be compensated for to the same degree. The right-hand side of Fig 5B shows the mean (over all examples in the test data set) absolute distance in standard deviations of the parameters of the simulation used to generate the posterior (i.e. red dots in Fig 5A) and the expectation of the posterior distribution. For all the parameters, the mean absolute distance is smaller than the standard deviation of the approximate marginal posterior distributions, indicating that the approximate posteriors generally covers the parameters of the simulation the posterior distribution is conditioned on.

One notable difference between the posterior distributions computed from the metamodel of the population spiking activities and the posterior distributions computed from the metamodel of the LFP is that the overall synaptic strength, $Q_s$, and the membrane capacitance, $C_m$, are significantly more constrained by observing the LFP than by the population spiking activities, with a standard deviation more than three times as large for the population spiking activities. This can also be observed in the example distributions shown in Fig 5A. For the rest of the parameters, the standard deviations of the 1D marginal distributions are roughly the same when computed for the population spiking activities and the LFP. The full posterior distribution from Fig 5A is shown in S1 Supplementary Section.

Fig 5C shows the metamodel output (orange) corresponding to the same simulation as in A, and 50 individual simulation outputs (black) run with parameter values sampled from the posterior distribution of the example shown in A. Only the spectra from the excitatory population spiking activity and the topmost LFP channel are shown. The mean of the metamodel distribution is shown in dashed orange, and the shaded area shows ± 2 standard deviations. The simulations run with parameters from the posterior distribution fit nicely with the distribution over the outputs given by the metamodel, suggesting that the posterior distribution is reasonable. Fig 5D shows the distribution of the maximum (taken over frequencies and simulations for a given posterior) distance between the simulation outputs run with parameters drawn from the posterior distributions, and the simulation output the posterior is conditioned on, for the 100 posterior distributions for which the extra simulations were run.

The black line shows the errors for the simulations run with parameters drawn from the posterior generated by the population spiking activities, while the gray distribution shows the distances for the simulations run with parameters drawn from the posterior generated by LFPs. The left-hand side shows the distances of the population spiking activities, while the right-hand side shows the distances for the LFP. For both the population spiking activities and the LFP, the simulations from the posterior distributions are very similar to the signal they were conditioned on. Simulations from the posterior distributions conditioned on the LFP, also give population spiking activities very similar to the ground-truth. Simulations from the posterior distributions conditioned on the population spiking activities, on the other hand, do not give as similar LFPs to the ground-truth, which again shows that observing the LFP constrains the parameters to a greater degree than observing the population spiking activities.

Fig 6 shows a comparison between the posteriors obtained by the DGPR metamodel and the MAF metamodel. Panel A shows the marginal standard deviations, averaged over all posterior distributions. For both metamodels, they are similar, but the DGPR is generally slightly wider. Panel B shows the same plot, but for the posterior predictive distribution, i.e. the distribution over outputs when parameters are sampled from the posterior distribution. The

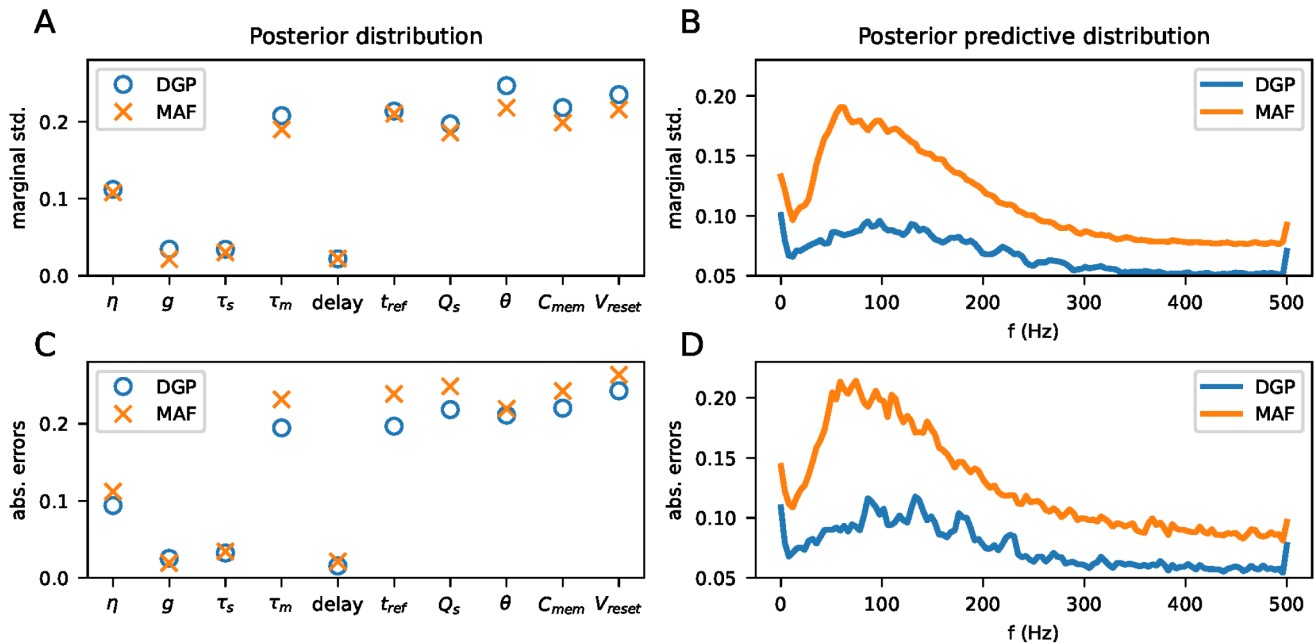

**Fig 6.** A: mean marginal standard deviations of the posterior distributions. B: mean marginal standard deviations of the posterior predictive distributions. C: mean distance between expectation of marginal posterior distribution and the parameters from the simulation the posterior distribution is conditioned on. D: Same as C, but for the posterior predictive distribution.

posterior predictive distributions from the DGPR metamodel are significantly narrower than those from the MAF metamodel, even with wider posterior distributions, indicating that the posterior distributions are more correct compared to those from the MAF metamodel. Panels C and D show the mean absolute distance between the expectation of the posterior distribution and posterior predictive distribution respectively, and the simulations they are conditioned on. For both models, the errors are on the same scale as the standard deviations, indicating that neither systematically misses the simulation it is conditioned on.

The Pearson correlation coefficient was computed for all pairwise combinations of parameters for all sampled posterior distributions. Fig 7 shows the mean of the correlation coefficients over all examples in the test data set. The upper triangle shows the coefficients for the samples from the posterior distribution conditioned on the population spiking activities, and the lower triangle shows the coefficients conditioned on the LFP. The correlations between most of the parameters are fairly weak, but since only the average is shown, there could be stronger correlations in individual posterior distributions that the plot does not capture. For the LFP, there are strong average positive correlations between the synaptic parameters, i.e. the synaptic strength $Q_s$, the synaptic time constant $\tau_{syn}$ and relative inhibitory strength $g$. Interestingly, the corresponding coefficients for the population spiking activities are close to 0. This again highlights the difference in the posterior distributions for $Q_s$ between the population spiking activities and the LFP. For the distributions conditioned on both the LFP and the population spiking activities, there are positive correlations are present between the parameters $\eta$ and $\tau_m$, as well as negative correlations between $\eta$, $V_{reset}$ and $C_m$. As these parameters all impact the amount of input required for a neuron to spike, there are possibly complex interactions between all these parameters yielding manifolds of similar activities and LFPs.

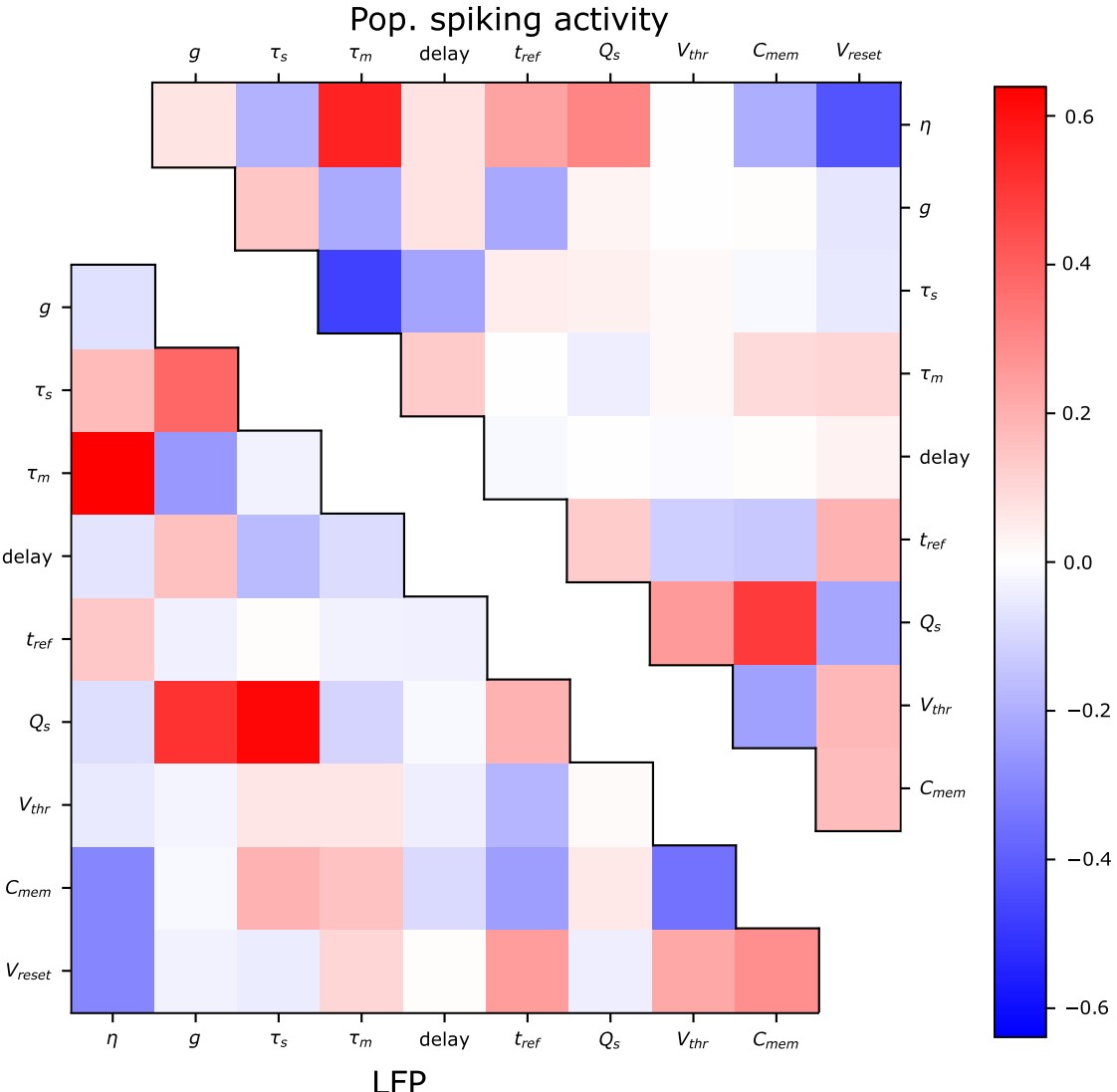

**Fig 7. Pearson correlation coefficients between samples from posterior distribution over parameters, averaged over the test data set.** Upper triangle shows the correlation coefficients for the posterior distributions from the simulated power spectra from the population spiking activities, while the lower triangle shows the same correlation coefficients from the LFP.

## 4 Discussion

In the present work we have trained metamodels to estimate the power spectra of population spiking activities and local field potentials (LFPs) generated by a two-population neuronal network comprising an excitatory and an inhibitory population during stationary activity. Specifically, we have trained a deep Gaussian process regression (DGPR) model, and a faster masked autoregressive flow (MAF) model, and used the probabilistic nature of the metamodels to estimate posterior distributions over parameters given observed simulation outputs. DGPR models have not been much used for statistical modelling in the field of neuroscience before, and we chose to compare it to the MAF as it is a commonly used model for both forward modelling in the case of SNLE and inverse modelling in the case of SNPE.

Both the population spiking activities and the LFP were modelled, as this offers an opportunity to compare the information about the parameters that can be gained by observing different signals. The LFP was computed separately for each population using Eq 10. This is an approximation that does not require computing the LFP response from each individual neuron in each simulation, which is computationally costly, but only requires computing a small number of kernels, which can then be used to approximate the LFP for all simulations. The synaptic parameters can be fully adjusted for by a single simulation of a kernel, but the neuron membrane conductance and capacitance, cannot be adjusted for analytically. Kernels were therefore computed on a grid of these two parameters, and regression was done to approximate the true kernel for any value of the membrane conductance and capacitance. As shown in S4 Supplementary Section, the approximation works well, although there are some small deviations, particularly in channels 4 and 5.

The neuron and synapse parameters of the simulations were varied, with values described in Table 2, while the number of neurons and number of synapses were fixed. The parameter values were chosen such that the strongly synchronous regions of the parameter space were avoided, particularly avoiding relative inhibitory synaptic strengths that lead to an unbalanced regime. This will also have excluded some AI-like simulations in the vicinity of the boundary between AI and SR, but we consider the difference (starting at $g = 4.5$ instead of $g = 4.0$) to be sufficiently small as to not significantly diminish the utility of the models. Additionally, a small number of simulations in the strongly synchronous irregular regime were removed, comprising roughly 3% of the total number of simulations as described in Sections 2.1 and 3.1. The simulations in the test and training data sets consisted mostly of simulations in the asynchronous irregular (AI) regime. The simulations with strongly synchronous spiking activities were avoided since sharp peaks that shift continuously in frequency cannot be well approximated by a low-dimensional representation, and the low-rank approximation was necessary to keep the size of the DGPR metamodel small. The low-rank approximation is a limiting factor for the DGPR metamodel, and ideally one would not have to limit the input domain in such a way. Increasing the dimensionality of the latent space significantly is possible and would possibly also have resulted in better accuracy, even in the case of strongly synchronous activity, particularly with a larger training data set, but trade-offs have to be made with respect to the computational cost of training and evaluating the metamodels. The simulations in the AI regime could be well approximated by a low-dimensional representation, and since one is often particularly interested in this regime, we consider it reasonable to focus the metamodels on this.

The power spectra were modelled instead of the time series. Although modelling time series directly in many cases would be preferable, that would require modelling the complex temporal dependencies in the neural networks, which are also stochastic due to the external input to the network modelled as a Poisson process. Since most regression techniques assume a fixed target, modelling the power spectra is more straightforward. For the present situation, we found that both the DGPR and MAF metamodels were able to accurately predict the power spectrum of the population spiking activities, with 10 different model parameters as inputs, and we found that on the order of around 1000 examples is sufficient to accurately represent the model in this particular region of the parameter space. In general, the number of examples needed to train a metamodel will depend on the number of parameters included, the range of their values and how much they affect the model output. It is difficult to predict a priori, and the presently used model may not be representative for other, more complex models, which may require more training examples.

Theoretical approaches based on linear-response theory can provide analytical approximations to the PSDs [42–45] in the asynchronous irregular regime. The theoretical models

necessarily make assumptions about the underlying physics of the system, which also makes it possible to interpret them in ways that are not possible with purely data-driven methods. Data-driven methods are complementary to theoretical ones in the sense that they do not make any underlying assumption about the model, but comes at the price of interpretability and needing to run simulations to train the models.

The forward modelling of the population spiking activities and the LFP is similar in terms of difficulty, where the errors of the DGPR metamodels are similar. This may not be surprising, as the LFP is computed as a convolution of the population spiking activities. The low-pass filtering effects of synaptic and dendritic filtering makes the power spectrum of the LFP smoother than the population spiking activities, however, which could potentially have made it an easier target [46, 47].

Comparing the DGPR metamodel to the MAF metamodel, we found that the DGPR metamodel provided better accuracy, but comes at the price of a higher cost of training. When it comes to running the trained models, the MAF is significantly faster than DGPR for evaluating the log probability, and therefore also when estimating posterior distributions. For purely sampling the forward model, the DGPR is faster than the MAF metamodel. This is not surprising, as the MAF model is designed to be good at evaluating log probabilities, at the cost of being slow at sampling. The time it takes both to train and evaluate the metamodels will depend strongly on the available hardware. GPU acceleration makes it significantly faster, and solely using CPUs would significantly increase the time of training/evaluating the metamodels.

The MAF metamodel suffered significantly more from overfitting compared to the DGPR metamodel. Simply increasing the number of simulations used to train the models would likely reduce the differences between the two metamodels. It is also possible that more regularization techniques could have increased the performance of the MAF, although utilizing dropout did not significantly improve the model.

For inverse modelling, the benefit of using the more computationally demanding DGPR will likely be smaller if one is doing sequential inference, and the region in which the metamodel must be accurate is smaller. Also depending on the complexity of the simulator one is approximating, in cases where one wants an accurate model over a larger domain, increased accuracy may be well worth the increased computational cost.

The HCPLSR metamodel, shown in S3 Supplementary Section, is significantly faster than both the DGPR and MAF metamodels, but at the cost of only giving a point-estimate of the simulation output, and less flexibility. For cases where one is only interested in the forward modelling, such alternatives can also be an option.

## 4.1 Model inversion

One key motivation for using probabilistic metamodels is that it allows inverting the model and finding posterior distributions over parameters given observed outputs. We opted to sample the posterior using an adaptive Metropolis algortihm as it is simple to implement and worked well for the present problem. The Gelman-Rubin statistic was computed, and for most of the sampled posterior distributions ($> 90\%$), it was below 1.1, which is often considered to indicate convergence [48]. For some of the posterior distributions, the chains did not converge, although this was likely only for a minority. Particularly for the examples on which the forward predictions are poor, we would not expect the estimated posterior distributions to be accurate, as the metamodel generally will not produce outputs which are similar to the observed output used to generate the posteriors.

The degree to which the different model parameters were constrained by observing an output varied highly (see Fig 5B). Notably the distribution over most neuron parameters were

wide, while the distribution over synapse and input parameters were narrower. Interestingly, the synaptic strength ($Q_s$) and membrane capacitance ($C_m$) were significantly more constrained by the LFP than by the population spiking activities, illustrating the utility of using different signals for model validation. The effect is likely due to the fact that LFP is directly dependent on both the post-synaptic currents as well as the population spiking activities. The effect was not seen for the synaptic time constant, which also directly influences the post-synaptic current, something that indicates that the extra information may be in the scale of the LFP, as that is directly determined by the scale of the post-synaptic currents. The remaining parameters were similarly constrained by both signals, indicating that for the present situation, both signals may contain similar information about the parameters.

A previous study on model inversion using the LFP of a similar network using convolutional neural network [30] provided more accurate predictions of parameter values. In that study, neuron models with delta synapses were used, and it was limited to only three parameters, which likely makes it an easier task. Particularly, wide marginal posterior distributions might only be possible if there are sufficiently many parameters that are free to vary such that they can compensate for the effects of other parameters. An interesting question which we did not pursue here is to what degree keeping the different parameters fixed constrains the remaining parameters given observed simulation outputs.

Comparing the posterior distributions generated from the DGPR metamodel to the posterior distribution generated from the MAF metamodel, we find that the posterior distributions from the DGPR metamodel are wider, while simultaneously having a significantly narrower posterior predictive distribution. In other words, it is able to find a larger region of the parameter space that gives more similar outputs compared to the MAF metamodel. This shows the importance of accurate forward modelling in order to find good posterior distributions, an area where DGPR metamodels can be useful also in other settings when dealing with data from complex simulations.

Methods such as Approximate Bayesian Computation (ABC) and Simulation-based Inference (SBI) can also be used to estimate posterior distributions over parameters given specific observed model outputs. Our work is closely related to SBI, particularly SNLE [13]. The approach in the present work can be considered the same as SNLE run for a single round, where we use the DGPR metamodel as a likelihood estimator. Thus, in cases where one is interested in estimating posterior distributions, particularly in cases where one has limited data and model flexibility is a concern, DGPR metamodels may be of use in the SNLE framework. Methods such as Sequential Neural Variational Inference [49] for estimating the posterior distribution instead of MCMC sampling can also readily be used with the DGPR metamodel.

## 4.2 Generalization to more complex network models

The two-population model is fairly simple, and is a useful starting point since the dynamics are well known, and the simulations are cheap to run. A natural question is how well these metamodelling techniques generalize to more complex networks such as a multi-population model of cortical circuitry [50], a multi-area model of the visual system [51, 52], or a large-scale model of mouse primary visual cortex [53]. Models with more populations will have more potential model outputs to capture, and richer dynamics, and a larger number of parameters will yield a higher dimensional input space. The expressiveness of the metamodels may have to increase, and the number of samples needed to train a metamodel will likely be much higher. Of course, the number of parameters one chooses to include is arbitrary, and if one is only interested in a subset of the parameters, that can be modelled more cheaply. It is

interesting to note, however, that in the present case, the power spectra mostly in the AI regime can be accurately captured by a low-dimensional latent representation.

In the present case, we model the external spiking input as a constant rate Poisson process, parameterised by the variable $\eta$. In more realistic networks, the input to the network will generally be more patterned. The parameters in the network we used were also kept constant, while in more realistic networks the parameters might follow a statistical distribution instead. Similar methods can be used to find the parameters of such distributions instead of the constant value as in the present case. Patterned input could also be represented in a metamodel, either through some continuous input to the model if the input changes in a continuous manner, or possibly discretely, in which case it may be necessary to train multiple metamodels for each input class separately.

The metamodels can easily be made more flexible, for the DGPR metamodel by increasing the number of GPs in either layer, and increasing the dimensionality in the latent space, and for the MAF metamodel by increasing the number of transformations or increasing the number of hidden features in each transformation. This will lead to higher computational costs, but more complex neural networks are of course also more costly to run, so the relative benefit may not change. In the present case, sampling was done using Latin Hypercube sampling, which ensures that each parameter is uniformly sampled. This can potentially lead to a too dense sampling in regions of the parameter space where the metamodel performs well, and too sparse sampling in regions where the metamodel performs poorly. Methods for adaptively sampling only or mostly in regions where additional sampling is needed would be an interesting direction for future research.

### 4.3 Use of metamodels

Once a metamodel has been trained, it can be evaluated much more rapidly than the model it is trained on. If it is well calibrated, it should be able to interpolate accurately inside the volume of the parameter space which was used to train it, and give a continuous approximation to the model behaviour. For models which are computationally demanding, trained metamodels could be distributed for use by others without requiring large computational resources.

For working with experimental data, metamodels could be useful in different ways. In the context of SBI, metamodels can be used as likelihood estimators in order to find parameters fitting network models to observed experimental data, in the same manner as in this work. Depending on which phenomena one is interested in modelling, this might require larger and more realistic network models than the one used here, but the principle would be the same. In the context of forward modelling, metamodels could be applied to experimental data to guide the choice of experimental parameters and their effects on experimental outcomes. How well this would work in practice, however, is not known. It would require training to specific experimental settings, and could be an interesting direction of future research.

### 4.4 Conclusion

We have shown that for the present situation, the power spectra of the population spiking activities and the local field potential can be accurately modelled by the DGPR and MAF metamodels, over a domain largely consisting of activity in the AI regime. The DGPR metamodel performed better than the MAF metamodel, but at the cost of higher computational demands for training and evaluation. The trained metamodels can be used to estimate posterior distributions over simulation parameters given observed simulation outputs. The posterior distribution estimated from the DGPR metamodel had higher marginal variances compared to the posterior distributions estimated from the MAF metamodel, while the posterior predictive

distributions from the DGPR metamodel were narrower compared to the posterior predictive distributions from the MAF metamodel, indicating more accurate posterior distributions. When dealing with complex models, utilizing DGPR for forward modelling may therefore be a good alternative, particularly when the amount of data is a limiting factor, and the increased training time is not a problem.

Based on the estimated posterior distributions, the LFP was found to contain significantly more information about the synaptic strength and membrane capacitance, while for the remaining eight parameters, both signals were equally informative.

## Supporting information

**S1 Supplementary Section. Hyperparameter scans for determining best metamodel configurations.**
(PDF)

**S2 Supplementary Section. Full posterior distributions from which the subset in Fig 5 is taken.**
(PDF)

**S3 Supplementary Section. Hierarchical Partial Least Squares Regression.** Description and results from HCPLSR metamodel.
(PDF)

**S4 Supplementary Section. Validation of LFP computation scheme.**
(PDF)

## Author Contributions

**Conceptualization:** Jan-Eirik W. Skaar, Alexander J. Stasik, Gaute T. Einevoll, Kristin Tøndel.

**Formal analysis:** Jan-Eirik W. Skaar, Nicolai Haug.

**Investigation:** Jan-Eirik W. Skaar.

**Methodology:** Jan-Eirik W. Skaar.

**Validation:** Jan-Eirik W. Skaar.

**Visualization:** Jan-Eirik W. Skaar.

**Writing – original draft:** Jan-Eirik W. Skaar.

**Writing – review & editing:** Jan-Eirik W. Skaar, Nicolai Haug, Gaute T. Einevoll, Kristin Tøndel.

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
