## [Decision Letter · Decision Letter 0]

19 Jan 2023

Dear Mr Skaar,

Thank you very much for submitting your manuscript "Metamodelling of a two-population spiking neural network" for consideration at PLOS Computational Biology.

As with all papers reviewed by the journal, your manuscript was reviewed by members of the editorial board and by several independent reviewers. In light of the reviews (below this email), we would like to invite the resubmission of a significantly-revised version that takes into account the reviewers' comments.

We cannot make any decision about publication until we have seen the revised manuscript and your response to the reviewers' comments. Your revised manuscript is also likely to be sent to reviewers for further evaluation.

Sincerely,

Tatiana Engel

Guest Editor

PLOS Computational Biology

Lyle Graham

Section Editor

PLOS Computational Biology

Reviewer's Responses to Questions

**Comments to the Authors:**

Reviewer #1: Skaar et al. present a thorough application of state-of-the-art metamodeling techniques to a highly-relevant and well-studied neural circuit model in theoretical neuroscience: leaky-IF spiking E/I ("Brunel") model. They accurately point out, that metamodeling -- the practice of approximating f(x | theta) or p(x | theta) for expensive simulation-defined models and then leveraging those approximations for scientific/theoretical inquiry -- is underutilized in the field of computational neuroscience. They explain the computational efficiency benefits of Hierarchical Clustering-based Partial Least Squares over alternative metamodeling approximation methods like deep gaussian process regression (GPR) and compare their application to the metamodeling of power spectral density of the Brunel model given 10 parameters (synaptic strengths, voltage, capacitance and time constant parameters of the model). Finally, they show that when executing simulation-based inference with their metamodels (as likelihoods) using MCMC, the spiking activities and LFPs result in different manifolds of degeneracy in the neural circuit model. I commend the authors on the degree of detail and clarity in the manuscript.

The work presented is a rigorous application of metamodeling techniques to the Brunel model, and advocates for a well-reasoned position that metamodeling techniques offer a practical, methodologically sound alternative to popular approaches in simulation-based inference. However, the findings presented do not reach the publication standard at PLOS Comp Bio.

- a.) Without showing any methodological comparison to prominent techniques like those referenced in the paper for simulation-based inference (SBI) (ABC sampling, SNPE, SNLE, etc), the work offers no evidence of a computational benefit from using either metamodeling (either GPR or HCPLS). We're also not given a sense of how the method scales with parametric dimensionality (important for neuroscience). After all, the ultimate reason for approximating a high-fidelity simulation-based neural circuit model would be to use that approximation for inversion/parameter inference. Otherwise, the contribution is that simulation cost is reduced by coarsening the fidelity of the neural circuit model simulation, which is not stated as a fundamental aim of this manuscript.

- b.) If this manuscript lacks methodological comparisons to SBI, then one would expect the methodology presented to at least generate some foundational insights about the neural circuit model being analyzed. The insights generated regarding the Brunel model are minimal, and only elaborated upon in descriptive terms. It is certainly interesting that the two types of conditioning yield different manifolds of degeneracy, but what does that mean about the model -- or about the physiological neurons they model. Can you make a novel prediction from this methodologically-generated insight and confirm it with model simulations?

I would be happy to re-read a manuscript, which has been re-written around a firm conclusion regarding lines a or b (or both).

Regarding line of work a.) a good place to start would be to insert the metamodel as an approximation of the simulation process in ABC and MCMC, and demonstrate the speed/accuracy trade-off. In what regimes (e.g. parameter count, simulation senarios (dt, NE, NI) etc.) of SBI does metamodeling offer computationally efficient, high-fidelity model inversion. To push the envelope, you could insert the metamodel into the relevant places of SNLE/SNPE (as you allude to) and do a similar analysis.

Reviewer #2: In this work, Skaar and colleagues construct metamodels of the classic Brunel spiking neural network to do fast emulation (i.e., simulate with the metamodel instead of the real simulator) and parameter posterior inference. They perform several thousand simulations of the Brunel network of excitatory and inhibitory populations with 10 free parameters drawn from a Latin hypercube prior, compute the power spectrum of the population rate, as well as PSD of the LFP, which is simulated in a separate hybrid-scheme model using just the spike times. The set of 4000 pairs of {network parameter, PSD} datapoints are used as training data for two types of metamodels, GP regression and hierarchical cluster-based PLS regression, to learn a fast emulator of the forward simulation. They compare the accuracy of the metamodel output against a held-out test set of simulations and find good performance for both, while GPR outperforms HCPLSR. With GPR, the authors then do model inversion to infer network parameters given an observed PSD (again, simulation from a held-out test set). They find reasonable parameter recovery performance and discover parameter correlations in the posterior distribution, while the posterior predictive simulations match that of the original ground-truth simulations. They derive insights about parameters of the Brunel network based on the inference results, such as the scaling between synaptic strength and time constant, which was dependent on which representation of the data (i.e., PSD of the pop rate vs. LFP) one uses as the observation, which in general affected the inferred posterior distribution, e.g., variance of the 1D marginal distribution for some parameters.

In general, I personally find this work very interesting and valuable, and is a great addition to the thread of works the group produces towards modeling / inverting network parameters from observed data. As far as I can tell, the work is rigorously done, though some clarifications can be made regarding hyperparam choices, and also the high-level motivation for model choices (i.e., HCPLSR, expanded below). One of the main concern I have is regarding using the hybrid scheme for LFP simulation with fixed parameters. In addition, the manuscript is a bit dense at points, and some visual presentation can be improved. I have a lot of comments below, which are only loosely ranked from high- to low-level comments (my apologies), but they are all meant to improve the clarity of the presentation or strengthen the arguments made in the paper.

- just to clarify, for the LFP simulations, are the hybrid / biophysical network parameters fixed for all simulations, i.e., do all LFP simulations use, for example, a membrane time constant of 20 ms, as implied in Table 4? If so, isn’t this problematic that the spiking network simulations are run with variable membrane time constants drawn from the prior, while the LFP is computed with a different and fixed time constant for all simulations? This may also explain why some parameters are more constrained when conditioned on the LFP spectra. Given that the spiking simulations are run anyway, wouldn’t it be straightforward to just compute the LFPs based on another approximation scheme (e.g., Mazzoni et al. 2015), which would presumably be a much better approximation of the LFP for each individual network. I think this is an important point to discuss, and possibly warrants further experiments, at least for a subset of the simulations, to show that the “true” LFP computed from currents in the spiking network (which have variable parameter values) match the LFP from the hybrid scheme (which has fixed parameter values), or that this does not influence posterior inference. Related to this, are PSDs from all 6 LFP channels used for inference? If not, the computational overhead of computing 1 LFP should be minimal?

- one thing that was not clear to me at all is the motivation behind using HCPLSR in this work. It is limited in expressivity while introducing computational overhead in the clustering step (as opposed to just doing nonlinear regression), does not allow model inversion, which is a main feature of the GP metamodel, and in general performs worse than the GP (though I suppose the latter point was only discovered by actually running the experiment). Given the group’s previous work (i.e., Skaar, Stasik et al 2020), I would have expected this comparison to be at least against a feedforward CNN model (or some kind of nonlinear regression model in general), which is probably equally fast for emulation time on modern computing hardware, but without loss of expressivity. In general, I think the results regarding HCPLSR is good to know, but does not at all add to the overall story, and can probably be moved to supplemental material given that the methods section is already quite technically challenging (which would also save Fig 2 to highlight a more important result). Alternatively, the authors can provide more motivation for trying HCPLSR (instead of a variety of other equivalent or superior regression techniques) , especially since they do not take advantage of the low dimensional representation found in PLS, as it would be interesting to see if the loadings for different clusters would correspond to qualitatively different regimes of network dynamics, etc.

- related to the previous point, it was emphasized in several places that the low dimensional representation in GP is a feature that affords more interpretability, but this was never taken advantage of in downstream analyses. I’m not saying this has to be added to this paper (though it would be very interesting to see!), but without actually looking into the GP latents, it undermines the key motivation for using it, which opens up the question of why not use a more expressive model. Similarly, given the previous results in Skaar, Stasik et al, it would have been natural to use a probabilistic model like mixture density network instead of a CNN to directly learn the posterior over parameters, with the advantage of bypassing the MCMC step as well as eliminating the utility of the GP’s smoothness. This is a rather general comment that can be addressed in writing of the introduction or discussion, or by performing additional analyses (of GP latents) or testing additional models (a more flexible replacement for the GP, or MDN for direct posterior estimation, etc.), and I leave it to the authors to address it in how they see fit.

- L121-131 implies that the prior bounds were restricted to eliminate parameter regions that result in the SR state, but wouldn’t that also exclude parameters that could generate AI dynamics? In the original Brunel phase diagram, this would correspond to the fact that the phase transition between SR and AI state has a slant in the 2D parameter space, while restricting the prior is akin to drawing a proper rectangle around only a part of the AI regime, and I would imagine this effect (i.e., boundaries not parallel to the coordinate axes in the parameter space) would only be exacerbated in the 10-D model here. A discussion of this would be warranted, esp since the authors attempt to capture a wide range of the model’s parameter space with the metamodel. A simple alternative would be to simulate from a broader prior, and post hoc throw out simulations in unwanted regimes (which the authors already do anyway).

- regarding the references to SBI methods (in the intro and discussion, L633-640), and in particular, SNPE/SNLE: while it’s true that the sequential version of these algorithms target posteriors for a specific observation, when run just for a single round, they have the exact same properties as the metamodels proposed here in that they are accurate over the full domain of the mechanistic model. In fact, SNLE run for a single round is exactly analogous to the GPR approach here, where the learned emulator/metamodel can be sampled from to produce fast simulations, but just replacing the gaussian process with another conditional density estimator, often of the neural network variety (e.g., normalizing flow). This fact does not diminish the novelty or how interesting the current work is, but is a point that should be clarified (and perhaps practical differences between estimators can then be further discussed, e.g., computational cost of GP vs. flows). Also, on L688: the metamodels plugged into SBI or ABC still produces a posterior that has to be MCMC sampled (as the authors do here), so I’m not sure what this statement is meant to convey?

- as a reader, the methodological description of GPR was extremely dense to read, while the technical details (and the current Figure 1) don’t contribute to the overall goal of the paper, especially since there’s no comparison of performance as a function of GP hyperparameters. I think most of this can be moved to supplemental materials, to make it easier for a non-technical reader to digest, though this is just my opinion. Similarly, I think the current figure 3 serves as a much better figure 1, as it provides an overview of the study, and should be highlighted as early as possible, instead of technical details about prior samples for the GP or schematic for PLSR. On that note, given the observation that the metamodel output PSD differ in smoothness from the actual simulations, maybe it’s worth exploring different length scales of the GP, as well as discussing the specific choice of hyperparameters (e.g., 10 and 14 GPs in layer 1 and 2, why?)

- some discussions of applying this approach to real neural data (and the limitations) may be of interest to a broader neuroscience audience

- does the latin hypercube prior outperform a simple uniform prior? Is there a practical or theoretical difference? This is just for my own curiosity.

- I think the first paragraph of section 3 (L327-334) is a fantastic summary of the work, and (some version of this) can be restated earlier in the paper as it really helped me as a reader to orient myself in the paper.

- are raw PSDs or log PSDs used as input for all the regression models (and at evaluation)? I’d imagine using the log PSD would help for the regression tasks given the 1/f.

- Fig 5D (and others): when evaluating metamodel output accuracy, I think it would be useful to scale the error to the range of the simulations from the prior samples (i.e., training set), or was this already done?

- I think some figures can be labeled more informatively:

- Fig 5A&B can have subtitles, pop rate and LFP PSD, respectively, same for the two rows of 5D; the lines/shaded area in 5A&B are in general very difficult to see; 5C x-axis can be labeled test set index, and perhaps could be plotted as a cumulative distribution (so the transpose of the current graph) to make the same point but with a more standard plot;

- Fig 8A can have subtitles denoting spiking and LFP (and the two columns in Fig 8 in general), and red dot/line can be labeled as ground-truth in the figure; it’s a bit confusing that Fig8B breaks the 2-column pattern (but not sure if much can be done about it); lines in 8C are very difficult to see, and I think the black lines entirely block the shaded orange? Is it an issue that the two columns of 8B look almost identical? Is there some straightforward explanation for this? Typo in caption for 8D (“orange dashed line in C”)

- I’m having a bit of a hard time understanding Fig 8D: is this the max distance of the metamodel mean from the posterior predictive simulations? Shouldn’t this be computed as a distance of the ground-truth simulation (i.e. the observation used for conditioning) from the posterior predictive simulations?

- L460: full 10D pairplot would be nice to see in supplemental

- L515-517: I think it would be clearer to use “conditioned on”, i.e., “…the coefficients for the samples from the posterior distribution conditioned on the population spiking activity,….and coefficients conditioned on the LFP”

Richard Gao, PhD

University of Tuebingen

Reviewer #3: The authors fit two statistical models ("metamodels"), to the Brunel network of excitatory and inhibitory LIF neurons. Some of the model parameters are treated as unknown and these unknown parameters are the input to the metamodel. The output of the metamodel are estimations of the power spectra of the population activities or of the LFP's. The metamodels are trained using supervised learning for many parameter sets from some domain of parameter space. The trained metamodels are tested on randomly sampled parameter combinations that lead to mostly asynchronous irregular spiking activity (parameters corresponding to (strong) synchronous activity are not considered both in the training and testing phase). The authors find an excellent performance of the metamodels both for the forward prediction of power spectra and for statistical inference of parameters given power spectrum estimates (inverse modeling). The parameter inference yields some interesting insights regarding the type of parameters and parameter combinations that can be well inferred, and the benefits of the inference on LFP data vs. population activity data (e.g. LFP data are advantageous for inference of synaptic connectivity). I found the paper important and interesting: the metamodel is a highly efficient model for the forward prediction of power spectra of LFPs or population activities given biophysically interpretable parameters, and inversely, the metamodel provides estimations of unknown biophysical parameters given power spectrum data. Both aspects are of high importance in neuroscience. The metamodelling framework is interesting because it combines the advantages of statistical models (efficiency, parameter inference) with the advantages of mechanistic network models (interpretability). The paper is scientifically sound and largely well written. The metamodel applies to state-of-the-art neural network models and LFP models in computational neuroscience and I expect the method to be highly useful at the interface of large-scale cortical recordings and network models of the corresponding circuits. The description of the method should be improved. Overall, I recommend publication in PLoS Computational Biology after the minor issues given below have been addressed.

Minor issues

1) The introduction of the concept of metamodels is rather abstract/general and would benefit from a more concrete description (alongside the general concept). Figure 3 is very nice and already quite helpful in this respect, but only comes in Sec.3. For example, it should be made clear already in the introduction that, in the present paper, the input of the metamodel are some parameters of the network model and the output of the metamodel is some statistics (e.g. power spectra or distribution over power spectra) of some functions of network variables (e.g. LFP or population activities). It could be stressed that the metamodel does not generate realizations of population activities or LFPs, in fact there is no "time" in the metamodel, in contrast to the network model and other statistical models.

2) Similar to point 1, I found the Methods part on Gaussian process regression rather abstract and incomprehensible to non-machine-learning experts. In Sec. 2.3, the quantities X, x and bold x seem to relate to the input, whereas the bold x has been defined as the output in the introduction (As I understand the bold x would be an estimate of the power spectra). Please, make sure that unique notations are used and all quantities are properly defined (x (not bold) does not seem to be defined, X should be defined more precisely, e.g. what is the dimension of the matrix, how is it constructed from n inputs (bold x?) of dimension d?). What is d in this paper? What is the "underlying function f(x)"? In line 200, if bold x denotes a vector (let's call it \\vec{x}), (\\vec{x}-\\vec{x}')^2 is not clearly defined. Do you mean the norm squared (dot product)?

3) What exactly is the output of the GPR? If it is the likelihood p(\\vec{x}|\\theta), how is it represented or computed (where is the equation for it)?

4) In line 269, the unnormalized posterior distribution is given in terms of the likelihood, which has been computed by the GPR. Why is it necessary to sample from this distribution? Isn't the unnormalized posterior distribution sufficient (e.g. to get the mean and standard deviation in the plots of this paper)?

5) Fig.5A,B: The 10 samples of the GPR (orange lines) are barely visible. On the other hand, I find it is not necessary and rather confusing to plot them. The output of the GPR is the distribution of the power spectrum, and hence it is sufficient to show the mean and the orange shaded area (two standard deviations). Therefore, I suggest to omit the orange lines, which will also lead to a better vivibility of the black lines. Furthermore, the authors write in line 385: "the smoothness of the power spectra is underestimated for lower frequencies, as can be observed by comparing the individual orange and black lines". I guess the mean power spectrum, which should be a much smoother function, is more relevant than individual samples because the mean power spectrum will provide an estimate of a perfectly averaged power spectrum in the network model (corresponding the averaging over infinitely many Hann windows in the Welch periodogram of an infinitely long simulation).

6) For me the "power spectrum" is synonymous with power spectral density (PSD), which is defined by dividing by the window length T: PSD(f)=<[Psi]^*(f)[Psi](f)>/T (otherwise the power spectrum would scale with T). Why is the power spectrum defined without division by T? The scipy.signal.welch also provides the PSD that is rescaled by the window size. The PSD has also the advantage that the firing rate of neurons can be easily read-off from the high-frequency limit of the PSD of population activities (apart from a factor N). The PSD is not dimensionless, e.g. the PSD of population rates has units of 1/s, whereas without division by T it is dimensionless.

7) What is the unit of the LFP in this paper? The units of the corresponding power spectra seem to be missing in the plots unless the LFP has units of 1/time and hence its Fourier transform is unitless.

8) It is unclear how the method works outside the asynchronous irregular regime and where it fails. Could the method be used also for the synchronous regime?

9) Are the rates nu in Fig.4 normalized by N? According to the definition of nu they are not, but in this case the single neuron rate would be nu/N which would result in extremely small rates. Furthermore, what exactly is nu here, is it a weighted sum of nu_E and nu_I? If nu is normalized then simply nu=nu_E=nu_I, but this is not the case if nu is not normalized.

10) There seems to be a numerical artifact in power spectrum at the highest frequency (strange decrease). This decrease should not be there because the PSD of the population activities saturates to a non-zero value (the saturation is already visible in the plots).

11) In the asynchronous irregular regime, linear-response theory (especially the theory based on Lindner Phys Rev E 72, 061919, 2005) usually provides highly accurate analytical approximations of the PSDs (see e.g. Trousdale et al. PLoS Comput Biol. 2012, Deger et al. Phys. Rev E 2014, and the simpler but less accurate theories of Brunel J. Comput Neurosci. 2000 and Bos, Helias et al. PLoS Comput Biol. 2016). These analytical formulas should provide an efficient alternative to metamodels, which should be discussed in the paper.

Typos

=====

-caption Fig.5: set, . -> set.

-caption Fig.5: errors are shown in standard deviations

-Table 1: shaped-shaped -> alpha-shaped?

-line 433: an an

-line 451: membrane potential tau_m -> membrane time constant tau_m

-caption Fig.8, last sentence: there should be two times a reference to panel C not B

-caption Fig.5: errors are shown in standard deviations -> maybe a bit better: errors are shown in units of the standard deviations

**Have the authors made all data and (if applicable) computational code underlying the findings in their manuscript fully available?**

Reviewer #1: Yes

Reviewer #2: Yes

Reviewer #3: Yes

PLOS authors have the option to publish the peer review history of their article (what does this mean?). If published, this will include your full peer review and any attached files.

Reviewer #1: **Yes: **Sean Bittner

Reviewer #2: **Yes: **Richard Gao

Reviewer #3: No
---

## [Decision Letter · Decision Letter 1]

27 Aug 2023

Dear Mr Skaar,

Thank you very much for submitting your manuscript "Metamodelling of a two-population spiking neural network" for consideration at PLOS Computational Biology. As with all papers reviewed by the journal, your manuscript was reviewed by members of the editorial board and by several independent reviewers. The reviewers appreciated the attention to an important topic. Based on the reviews, we are likely to accept this manuscript for publication, providing that you address the concerns raised by the reviewer in the revised manuscript. In particular, state clearly what advances your work provides and what is the significance of your study for the computational neuroscience field.

Sincerely,

Tatiana Engel

Guest Editor

PLOS Computational Biology

Lyle Graham

Section Editor

PLOS Computational Biology

Reviewer's Responses to Questions

**Comments to the Authors:**

Reviewer #1: Recommendation

The authors have made diligent efforts to account for feedback in the previous round, which they should be commended for. This manuscript now very clearly lays out the role of metamodeling with respect to state-of-the-art techniques used in computational neuroscience for SBI - namely the normalizing flows (MAFs) used to estimate likelihoods (forward model) for inference as in SNLE. SNPE is primarily used in favor of SNLE, so I find this paper’s focus on the metamodel (estimation of the likelihood/forward model) to be a refreshing re-examination of the utility of such approaches for models with expensive simulations with intricate emergent phenomena like the Brunel model. However, I struggle to identify an aspect of the advancements/insights laid out in this paper that meet the publication criteria of PLoS Computational Biology. I recommend rejecting this paper for the reasons elaborated below.

Novelty/innovation/insight

In the first review, I stated my opinion that this manuscript could meet the publication criteria for this journal along the lines of innovation/insight if they did either one of two things: a.) demonstrate benefits of metamodeling with respect to SoTA approaches from of SBI or b.) generate foundational insights about the Brunel model using these techniques. Showing that a metamodel can be used to approximate the Brunel model to a reasonable degree (which they have definitely done) is not enough. While they have made considerable progress towards each of these aims, they don’t present enough evidence of a.) methodological advancement or enough analyses to yield b.) foundational insight generated with this methodology.

a.) methodological advancement

Since the original manuscript had a focus on using metamodels towards parameter inference, the authors followed recommendations from Reviewers 1 and 2 to explore the methodological tradeoffs between metamodeling and methods in SBI. Since metamodeling (estimation of the likelihood) is a process within SNLE, the authors accurately point out that the process undertaken in the manuscript is like running SNLE for 1 round. This underscores the fact that metamodeling isn’t novel in computational neuroscience, however it is certainly underutilized, and they will be making a case for its use in this paper.

The methodological comparisons in the latest manuscript focus on the approximating model of the Brunel model - the type of metamodel. Indeed they show some evidence that deep gaussian processes are a better approximation than MAFs. for the Brunel model, and also that HCPLSR lacks the expressivity necessary for this problem (I appreciate the inclusion of this in the manuscript knowing how great a focus this was in the original version). Even if this claim that DGPR was a more effective method of metamodeling than MAF for the Brunel model was firmly established (which I'm not entirely sure it is, see section Rigor/completeness), this relative advantage of DGPR would need to be demonstrated for at least two neural circuit models (not just one) to establish that it is in fact a better practice for the field moving forward – thus constituting an advancement worthy of publication.

After this first round of revision, I would have expected the authors to take a more focused approach on simulation count/computational efficiency – what is the trade-off space (simulation count and/or computational cost vs inference/estimation quality) of HCPLSR or DGPR metamodels vs established methods in SBI.

b.) foundational insight

The authors certainly made a more concerted effort to expand on the meaning of the posteriors inferred via the metamodel, however I don’t believe any concrete insights have been revealed in this manuscript. There are certainly paragraphs in Section 3.4 and Discussion along these lines, but they don’t rise beyond descriptions of the visualized/quantified correlations in the posterior, or 1D marginal variances, with some prose about how these parameters relate in an information theoretic context.

To meet publication criteria, we'd want to gain some interpretable insight from inferred posteriors - stipulate a hypothesis(es) from this insight - and then confirm them in simulation. That is not done here.

The fact that conditioning on LFPs vs spiking activity yields less posterior variance is nice to see, but I’m not sure this is surprising or has generated any new understanding of the Brunel model and its function.

Rigor/completeness

Some of this manuscript is well written and mature, but I found several parts unclear.

- Figure 3 - No mention or explanation is given for why MAF is missing from the right side of the figure.

- How many parameters are used in ideal DGPR model vs ideal MAF model? Can this explain the overfitting mentioned in the main text?

- Figure 4 - train and test labels are switched (otherwise something is very wrong).

- Are these training errors evaluated on the progressively increasing training set, or the complete dataset irrespective of the amount used in training?

- How many simulations were used to build the metamodels used in the following analyses? 1000 because its the elbow you designated?

- What’s the relative computational cost (training time per epoch etc.) of fitting these metamodels to simulation cost? I don’t have an idea of what the true bottleneck in this parameter inference problem is given the methods you’re exploring.

- Figure 5 - Which metamodel was used in this figure? I assume DGPR, but it’s not explicitly stated.

- Figure 5B - no explanation of what x and o are in this figure.

- Figure 5D should have x-axis labels indicating the difference of left vs right.

- Figure 6

- I agree that Figure 6D using the mean (or median) error of the posterior predictive wrt to the conditioning simulation is an appropriate evaluation method for the inference techniques shown here, but I disagree with the interpretation of Figure 6B, and the statement L466-469 about the posterior predictive variance.

- The posterior predictive distribution will reflect the uncertainty of the approximate posterior. A method that underestimates the posterior variance, will have lower posterior predictive variance, and that certainly doesn’t make it a better method than one accurately capturing the posterior variance.

Reviewer #2: Thanks to the authors for constructively engaging with my comments. The new experiments, especially the comparison with MAF, are quite interesting and now an additional contribution the paper makes. I'm happy with the manuscript as it is and very excited to see it in publication.

Reviewer #3: The authors have satisfactorily addressed all my comments.

**Have the authors made all data and (if applicable) computational code underlying the findings in their manuscript fully available?**

Reviewer #1: Yes

Reviewer #2: Yes

Reviewer #3: None

PLOS authors have the option to publish the peer review history of their article (what does this mean?). If published, this will include your full peer review and any attached files.

Reviewer #1: **Yes: **Sean Bittner

Reviewer #2: **Yes: **Richard Gao

Reviewer #3: **Yes: **Tilo Schwalger

Figure Files:

Data Requirements:

Reproducibility:

References:

---

## [Decision Letter · Decision Letter 2]

23 Oct 2023

Dear Mr Skaar,

We are pleased to inform you that your manuscript 'Metamodelling of a two-population spiking neural network' has been provisionally accepted for publication in PLOS Computational Biology.

Best regards,

Tatiana Engel

Guest Editor

PLOS Computational Biology

Lyle Graham

Section Editor

PLOS Computational Biology

Reviewer's Responses to Questions

**Comments to the Authors:**

Reviewer #1: My concerns from the previous round of review have been addressed, and I'm happy to endorse the acceptance of this manuscript. This is a great paper that will add an important complementary perspective to the emerging field of methodological techniques for handling expensive simulation-based models in theoretical neuroscience.

**Have the authors made all data and (if applicable) computational code underlying the findings in their manuscript fully available?**

Reviewer #1: None

PLOS authors have the option to publish the peer review history of their article (what does this mean?). If published, this will include your full peer review and any attached files.

Reviewer #1: **Yes: **Sean Bittner

---

## [Editor Report · Acceptance letter]

3 Nov 2023

PCOMPBIOL-D-22-01359R2 

Metamodelling of a two-population spiking neural network

Dear Dr Skaar,

I am pleased to inform you that your manuscript has been formally accepted for publication in PLOS Computational Biology. Your manuscript is now with our production department and you will be notified of the publication date in due course.

With kind regards,

Zsofi Zombor
